# Scalable Adaptation of State Complexity for Nonparametric Hidden Markov Models

**Michael C. Hughes, William Stephenson, and Erik B. Sudderth**
Department of Computer Science, Brown University, Providence, RI 02912
mhughes@cs.brown.edu, wtstephe@gmail.com, sudderth@cs.brown.edu

## Abstract

Bayesian nonparametric hidden Markov models are typically learned via fixed truncations of the infinite state space or local Monte Carlo proposals that make small changes to the state space. We develop an inference algorithm for the sticky hierarchical Dirichlet process hidden Markov model that scales to big datasets by processing a few sequences at a time yet allows rapid adaptation of the state space cardinality. Unlike previous point-estimate methods, our novel variational bound penalizes redundant or irrelevant states and thus enables optimization of the state space. Our birth proposals use observed data statistics to create useful new states that escape local optima. Merge and delete proposals remove ineffective states to yield simpler models with more affordable future computations. Experiments on speaker diarization, motion capture, and epigenetic chromatin datasets discover models that are more compact, more interpretable, and better aligned to ground truth segmentations than competitors. We have released an open-source Python implementation which can parallelize local inference steps across sequences.

## 1 Introduction

The *hidden Markov model* (HMM) [1, 2] is widely used to segment sequential data into interpretable discrete states. Human activity streams might use walking or dancing states, while DNA transcription might be understood via promotor or repressor states [3]. The *hierarchical Dirichlet process HMM* (HDP-HMM) [4, 5, 6] provides an elegant Bayesian nonparametric framework for reasoning about possible data segmentations with different numbers of states.

Existing inference algorithms for HMMs and HDP-HMMs have numerous shortcomings: they cannot efficiently learn from large datasets, do not effectively explore segmentations with varying numbers of states, and are often trapped at local optima near their initialization. Stochastic optimization methods [7, 8] are particularly vulnerable to these last two issues, since they cannot change the number of states instantiated during execution. The importance of removing irrelevant states has been long recognized [9]. Samplers that add or remove states via split and merge moves have been developed for HDP topic models [10, 11] and beta process HMMs [12]. However, these Monte Carlo proposals use the entire dataset and require all sequences to fit in memory, limiting scalability.

We propose an HDP-HMM learning algorithm that reliably transforms an uninformative, single-state initialization into an accurate yet compact set of states. Generalizing previous work on *memoized* variational inference for DP mixture models [13] and HDP topic models [14], we derive a variational bound for the HDP-HMM that accounts for sticky state persistence *and* can be used for effective Bayesian model selection. Our algorithm uses *birth* proposal moves to create new states and *merge* and *delete* moves to remove states with poor predictive power. State space adaptations are validated via a global variational bound, but by caching sufficient statistics our memoized algorithm efficiently processes subsets of sequences at each step. Extensive experiments demonstrate the reliability and scalability of our approach, which can be reproduced via Python code we have released online[1].

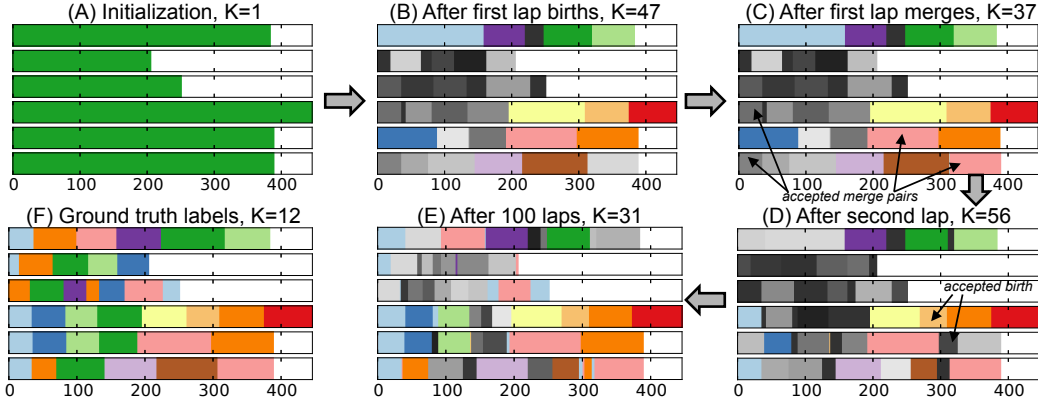

Figure 1: Illustration of our new birth/merge/delete variational algorithm as it learns to segment motion capture sequences into common exercise types (Sec. 5). Each panel shows segmentations of the same 6 sequences, with time on the horizontal axis. Starting from just one state (A), birth moves at the first sequence create useful states. Local updates to each sequence in turn can use existing states or birth new ones (B). After all sequences are updated once, we perform merge moves to clean up and *lap* is complete (C). After another complete lap of birth updates at each sequence followed by merges and deletes, the segmentation is further refined (D). After many laps, our final segmentation (E) aligns well to labels from a human annotator (F), with some true states aligning to multiple learned states that capture subject-specific variability in exercises.

## 2  Hierarchical Dirichlet Process Hidden Markov Models

We wish to jointly model $N$ sequences, where sequence $n$ has data $x_n = [x_{n1}, x_{n2}, \ldots, x_{nT_n}]$ and observation $x_{nt}$ is a vector representing interval or timestep $t$. For example, $x_{nt} \in \mathbb{R}^D$ could be the spectrogram for an instant of audio, or human limb positions during a 100ms interval.

The HDP-HMM explains this data by assigning each observation $x_{nt}$ to a single hidden state $z_{nt}$. The chosen state comes from a countably infinite set of options $k \in \{1, 2, \ldots\}$, generated via Markovian dynamics with initial state distributions $\pi_0$ and transition distributions $\{\pi_k\}_{k=1}^{\infty}$:

$$p(z_{n1} = k) = \pi_{0k}, \quad p(z_{nt} = \ell \mid z_{n,t-1} = k) = \pi_{k\ell}. \tag{1}$$

We draw data $x_{nt}$ given assigned state $z_{nt} = k$ from an exponential family likelihood $F$:

$$F : \log p(x_{nt} \mid \phi_k) = s_F(x_{dn})^T \phi_k + c_F(\phi_k), \qquad H : \log p(\phi_k \mid \bar{\tau}) = \phi_k^T \bar{\tau} + c_H(\bar{\tau}). \tag{2}$$

The natural parameter $\phi_k$ for each state has conjugate prior $H$. Cumulant functions $c_F, c_H$ ensure these distributions are normalized. The chosen exponential family is defined by its sufficient statistics $s_F$. Our experiments consider Bernoulli, Gaussian, and auto-regressive Gaussian likelihoods.

**Hierarchies of Dirichlet processes.** Under the HDP-HMM prior and posterior, the number of states is unbounded; it is possible that every observation comes from a unique state. The *hierarchical Dirichlet process* (HDP) [5] encourages sharing states over time via a latent root probability vector $\beta$ over the infinite set of states (see Fig. 2). The *stick-breaking representation* of the prior on $\beta$ first draws independent variables $u_k \sim \text{Beta}(1, \gamma)$ for each state $k$, and then sets $\beta_k = u_k \prod_{\ell=1}^{k-1}(1 - u_\ell)$. We interpret $u_k$ as the conditional probability of choosing state $k$ among states $\{k, k+1, k+2, \ldots\}$.

In expectation, the $K$ most common states are first in stick-breaking order. We represent their probabilities via the vector $[\beta_1 \ \beta_2 \ \ldots \ \beta_K \ \beta_{>K}]$, where $\beta_{>K} = \sum_{k=K+1}^{\infty} \beta_k$. Given this $(K+1)$-dimensional probability vector $\beta$, the HDP-HMM generates transition distributions $\pi_k$ for each state $k$ from a Dirichlet with mean equal to $\beta$ and variance governed by concentration parameter $\alpha > 0$:

$$[\pi_{k1} \ \ldots \ \pi_{kK} \ \pi_{k>K}] \sim \text{Dir}(\alpha\beta_1, \alpha\beta_2, \ldots, \alpha\beta_{>K}). \tag{3}$$

We draw starting probability vector $\pi_0$ from a similar prior with much smaller variance, $\pi_0 \sim \text{Dir}(\alpha_0\beta)$ with $\alpha_0 \gg \alpha$, because few starting states are observed.

**Sticky self-transition bias.** In many applications, we expect each segment to persist for many timesteps. The "sticky" parameterization of [4, 6] favors self-transition by placing extra prior mass on the transition probability $\pi_{kk}$. In particular, $[\pi_{k1} \ \ldots \ \pi_{k>K}] \sim \text{Dir}(\alpha\beta_1, \ldots \alpha\beta_k + \kappa, \ldots \alpha\beta_{>K})$ where $\kappa > 0$ controls the degree of self-transition bias. Choosing $\kappa \approx 100$ leads to long segment lengths, while avoiding the computational cost of semi-Markov alternatives [7].

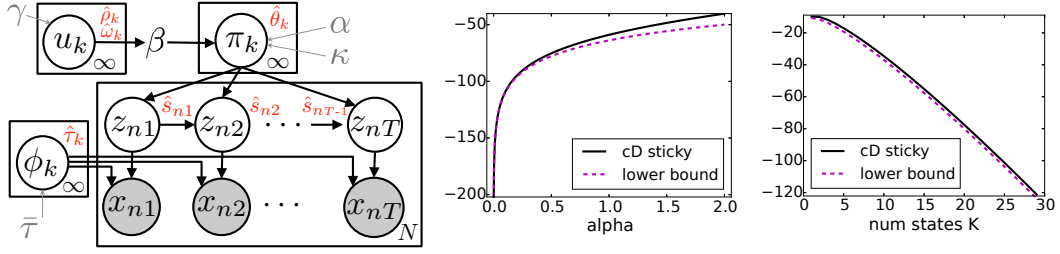

Figure 2: *Left:* Graphical representation of the HDP hidden Markov model. Variational parameters are shown in red. *Center:* Our surrogate bound for the sticky Dirichlet cumulant function $c_D$ (Eq. 9) as a function of $\alpha$, computed with $\kappa = 100$ and uniform $\beta$ with $K = 20$ active states. *Right:* Surrogate bound vs. $K$, with fixed $\kappa = 100, \alpha = 0.5$. This bound remains tight when our state adaptation moves insert or remove states.

## 3 Memoized and Stochastic Variational Inference

After observing data $x$, our inferential goal is posterior knowledge of top-level conditional probabilities $u$, HMM parameters $\pi, \phi$, and assignments $z$. We refer to $u, \pi, \phi$ as *global* parameters because they generalize to new data sequences. In contrast, the states $z_n$ are *local* to a specific sequence $x_n$.

### 3.1 A Factorized Variational Lower Bound

We seek a distribution $q$ over the unobserved variables that is close to the true posterior, but lies in the simpler factorized family $q(\cdot) \triangleq q(u)q(\phi)q(\pi)q(z)$. Each factor has exponential family form with free parameters denoted by hats, and our inference algorithms update these parameters to minimize the Kullback-Leibler (KL) divergence $\mathrm{KL}(q \parallel p)$. Our chosen factorization for $q$ is similar to [7], but includes a substantially more accurate approximation to $q(u)$ as detailed in Sec. 3.2.

**Factor** $q(z)$. For each sequence $n$, we use an independent factor $q(z_n)$ with Markovian structure:

$$q(z_n) \triangleq \left[ \prod_{k=1}^{K} \hat{r}_{n1k}^{\delta_k(z_{n1})} \right] \prod_{t=1}^{T-1} \prod_{k=1}^{K} \prod_{\ell=1}^{K} \left[ \frac{\hat{s}_{ntk\ell}}{\hat{r}_{ntk}} \right]^{\delta_k(z_{nt})\delta_\ell(z_{n,t+1})} \tag{4}$$

Free parameter vector $\hat{s}_{nt}$ defines the joint assignment probabilities $\hat{s}_{ntk\ell} \triangleq q(z_{n,t+1} = \ell, z_{nt} = k)$, so the $K^2$ non-negative entries of $\hat{s}_{nt}$ sum to one. The parameter $\hat{r}_{nt}$ defines the marginal probability $\hat{r}_{ntk} = q(z_{nt} = k)$, and equals $\hat{r}_{ntk} = \sum_{\ell=1}^{K} \hat{s}_{ntk\ell}$. We can find the expected count of transitions from state $k$ to $\ell$ across all sequences via the sufficient statistic $M_{k\ell}(\hat{s}) \triangleq \sum_{n=1}^{N} \sum_{t=1}^{T_n-1} \hat{s}_{ntk\ell}$.

The truncation level $K$ limits the total number of states to which data is assigned. Under our approximate posterior, only $q(z_n)$ is constrained by this choice; no global factors are truncated. Indeed, if data is only assigned to the first $K$ states, the conditional independence properties of the HDP-HMM imply that $\{\phi_k, u_k \mid k > K\}$ are independent of the data. Their optimal variational posteriors thus match the prior, and need not be explicitly computed or stored [15, 16]. Simple variational algorithms treat $K$ as a fixed constant [7], but Sec. 4 develops novel algorithms that fit $K$ to data.

**Factor** $q(\pi)$. For the starting state ($k = 0$) and each state $k \in 1, 2, \ldots$, we define $q(\pi_k)$ as a Dirichlet distribution: $q(\pi_k) \triangleq \mathrm{Dir}(\hat{\theta}_{k1}, \ldots, \hat{\theta}_{kK}, \hat{\theta}_{k>K})$. Free parameter $\hat{\theta}_k$ is a vector of $K + 1$ positive numbers, with one entry for each of the $K$ active states and a final entry for the aggregate mass of all other states. The expected log transition probability between states $k$ and $\ell$, $P_{k\ell}(\hat{\theta}) \triangleq \mathbb{E}_q[\log \pi_{k\ell}] = \psi(\hat{\theta}_{k\ell}) - \psi(\sum_{m=1}^{K+1} \hat{\theta}_{km})$, is a key sufficient statistic.

**Factor** $q(\phi)$. Emission parameter $\phi_k$ for state $k$ has factor $q(\phi_k) \triangleq H(\hat{\tau}_k)$ conjugate to the likelihood $F$. The supplement provides details for Bernoulli, Gaussian, and auto-regressive $F$.

We score the approximation $q$ via an objective function $\mathcal{L}$ that assigns a scalar value (higher is better) to each possible input of free parameters, data $x$, and hyperparameters $\gamma, \alpha, \kappa, \bar{\tau}$:

$$\mathcal{L}(\cdot) \triangleq \mathbb{E}_q\left[\log p(x, z, \pi, u, \phi) - \log q(z, \pi, u, \phi)\right] = \mathcal{L}_{\text{data}} + \mathcal{L}_{\text{entropy}} + \mathcal{L}_{\text{hdp-local}} + \mathcal{L}_{\text{hdp-global}}. \tag{5}$$

This function provides a lower bound on the marginal evidence: $\log p(x|\gamma, \alpha, \kappa, \bar{\tau}) \geq \mathcal{L}$. Improving this bound is equivalent to minimizing $\text{KL}(q \parallel p)$. Its four component terms are defined as follows:

$$\mathcal{L}_{\text{data}}(x, \hat{r}, \hat{\tau}) \triangleq \mathbb{E}_q\left[\log p(x \mid z, \phi) + \log \frac{p(\phi)}{q(\phi)}\right], \qquad \mathcal{L}_{\text{entropy}}(\hat{s}) \triangleq -\mathbb{E}_q\left[\log q(z)\right],$$

$$\mathcal{L}_{\text{hdp-local}}(\hat{s}, \hat{\theta}, \hat{\rho}, \hat{\omega}) \triangleq \mathbb{E}_q\left[\log p(z \mid \pi) + \log \frac{p(\pi)}{q(\pi)}\right], \qquad \mathcal{L}_{\text{hdp-global}}(\hat{\rho}, \hat{\omega}) \triangleq \mathbb{E}_q\left[\log \frac{p(u)}{q(u)}\right]. \tag{6}$$

Detailed analytic expansions for each term are available in the supplement.

## 3.2 Tractable Posterior Inference for Global State Probabilities

Previous variational methods for the HDP-HMM [7], and for HDP topic models [16] and HDP grammars [17], used a zero-variance point estimate for the top-level state probabilities $\beta$. While this approximation simplifies inference, the variational objective no longer bounds the marginal evidence. Such pseudo-bounds are unsuitable for model selection and can favor models with redundant states that do not explain any data, but nevertheless increase computational and storage costs [14].

Because we seek to learn compact and interpretable models, and automatically adapt the truncation level $K$ to each dataset, we instead place a proper beta distribution on $u_k$, $k \in 1, 2, \ldots K$:

$$q(u_k) \triangleq \text{Beta}(\hat{\rho}_k \hat{\omega}_k, (1-\hat{\rho}_k)\hat{\omega}_k), \text{ where } \hat{\rho}_k \in (0, 1), \hat{\omega}_k > 0. \tag{7}$$

Here $\hat{\rho}_k = \mathbb{E}_{q(u)}[u_k]$, $\mathbb{E}_{q(u)}[\beta_k] = \hat{\rho}_k \mathbb{E}[\beta_{>k-1}]$, and $\mathbb{E}_{q(u)}[\beta_{>k}] = \prod_{\ell=1}^{k}(1-\hat{\rho}_\ell)$. The scalar $\hat{\omega}_k$ controls the variance, where the zero-variance point estimate is recovered as $\hat{\omega}_k \to \infty$.

The beta factorization in Eq. (7) complicates evaluation of the marginal likelihood bound in Eq. (6):

$$\mathcal{L}_{\text{hdp-local}}(\hat{s}, \hat{\theta}, \hat{\rho}, \hat{\omega}) = \mathbb{E}_{q(u)}[c_D(\alpha_0 \beta)] + \sum_{k=1}^{K} \mathbb{E}_{q(u)}[c_D(\alpha\beta + \kappa\delta_k)]$$
$$- \sum_{k=0}^{K} c_D(\hat{\theta}_k) + \sum_{k=0}^{K} \sum_{\ell=1}^{K+1} (M_{k\ell}(\hat{s}) + \alpha_k \mathbb{E}_{q(u)}[\beta_\ell] + \kappa\delta_k(\ell) - \hat{\theta}_{k\ell}) P_{k\ell}(\hat{\theta}). \tag{8}$$

The Dirichlet cumulant function $c_D$ maps $K+1$ positive parameters to a log-normalization constant. For a non-sticky HDP-HMM where $\kappa = 0$, previous work [14] established the following bound:

$$c_D(\alpha\beta) \triangleq \log \Gamma(\alpha) - \sum_{k=1}^{K+1} \log \Gamma(\alpha\beta_k) \geq K \log \alpha + \sum_{\ell=1}^{K+1} \log \beta_\ell. \tag{9}$$

Direct evaluation of $\mathbb{E}_{q(u)}[c_D(\alpha\beta)]$ is problematic because the expectations of log-gamma functions have no closed form, but the lower bound has a simple expectation given beta distributed $q(u_k)$.

Developing a similar bound for sticky models with $\kappa > 0$ requires a novel contribution. To begin, in the supplement we establish the following bound for any $\kappa > 0, \alpha > 0$:

$$c_D(\alpha\beta + \kappa\delta_k) \geq K \log \alpha - \log(\alpha + \kappa) + \log(\alpha\beta_k + \kappa) + \sum_{\ell=1 \; \ell\neq k}^{K+1} \log(\beta_\ell). \tag{10}$$

To handle the intractable term $\mathbb{E}_{q(u)}[\log(\alpha\beta_k + \kappa)]$, we leverage the concavity of the logarithm:

$$\log(\alpha\beta_k + \kappa) \geq \beta_k \log(\alpha + \kappa) + (1 - \beta_k) \log \kappa. \tag{11}$$

Combining Eqs. (10) and (11) and taking expectations, we can evaluate a lower bound on Eq. (8) in closed form, and thereby efficiently optimize its parameters. As illustrated in Fig. 2, this rigorous lower bound on the marginal evidence $\log p(x)$ is quite accurate for practical hyperparameters.

## 3.3 Batch and Stochastic Variational Inference

Most variational inference algorithms maximize $\mathcal{L}$ via coordinate ascent optimization, where the best value of each parameter is found given fixed values for other variational factors. For the HDP-HMM this leads to the following updates, which when iterated converge to some local maximum.

**Local update to $q(z_n)$.** The assignments for each sequence $z_n$ can be updated independently via dynamic programming [18]. The forward-backward algorithm takes as input a $T_n \times K$ matrix of log-likelihoods $\mathbb{E}_q[\log p(x_n \mid \phi_k)]$ given the current $\hat{\tau}$, and log transition probabilities $P_{jk}$ given the current $\hat{\theta}$. It outputs the optimal marginal state probabilities $\hat{s}_n, \hat{r}_n$ under objective $\mathcal{L}$. This step has cost $\mathcal{O}(T_n K^2)$ for sequence $n$, and we can process multiple sequences in parallel for efficiency.

**Global update to $q(\phi)$.** Conjugate priors lead to simple closed-form updates $\hat{\tau}_k = \bar{\tau} + S_k$, where sufficient statistic $S_k$ summarizes the data assigned to state $k$: $S_k \triangleq \sum_{n=1}^{N} \sum_{t=1}^{T_n} \hat{r}_{ntk} s_F(x_{nt})$.

**Global update to $q(\pi)$.** For each state $k \in \{0, 1, 2 \ldots K\}$, the positive vector $\hat{\theta}_k$ defining the optimal Dirichlet posterior on transition probabilities from state $k$ is $\hat{\theta}_{k\ell} = M_{k\ell}(\hat{s}) + \alpha\beta_\ell + \kappa\delta_k(\ell)$. Statistic $M_{k\ell}(\hat{s})$ counts the expected number of transitions from state $k$ to $\ell$ across all sequences.

**Global update to** $q(u)$**.** Due to non-conjugacy, our surrogate objective $\mathcal{L}$ has no closed-form update to $q(u)$. Instead, we employ numerical optimization to update vectors $\hat{\rho}, \hat{\omega}$ simultaneously:

$$\arg\max_{\hat{\rho}, \hat{\omega}} \mathcal{L}_{\text{hdp-local}}(\hat{\rho}, \hat{\omega}, \hat{\theta}, \hat{s}) + \mathcal{L}_{\text{hdp-global}}(\hat{\rho}, \hat{\omega}) \quad \text{subject to } \hat{\omega}_k > 0, \hat{\rho}_k \in (0, 1) \text{ for } k = 1, 2 \dots K.$$

Details are in the supplement. The update to $q(u)$ requires expectations under $q(\pi)$, and vice versa, so it can be useful to iteratively optimize $q(\pi)$ and $q(u)$ several times given fixed local statistics.

To handle large datasets, we can adapt these updates to perform *stochastic variational inference* (SVI) [19]. Stochastic algorithms perform local updates on random subsets of sequences (batches), and then perturb global parameters by following a noisy estimate of the natural gradient, which has a simple closed form. SVI has previously been applied to non-sticky HDP-HMMs with point-estimated $\beta$ [7], and can be easily adapted to our more principled objective. One drawback of SVI is the requirement of a learning rate schedule, which must typically be tuned to each dataset.

### 3.4 Memoized Variational Inference

We now outline a memoized algorithm [13] for our sticky HDP-HMM variational objective. Before execution, each sequence is randomly assigned to one of $B$ batches. The algorithm repeatedly visits batches one at a time in random order; we call each full pass through the complete set of $B$ batches a *lap*. At each visit to batch $b$, we perform a local step for all sequences $n$ in batch $b$ and then a global step. With $B = 1$ batches, memoized inference reduces to the standard full-dataset algorithm, while with larger $B$ we have more affordable local steps and faster overall convergence. With just one lap, memoized inference is equivalent to the synchronous version of *streaming variational inference*, presented in Alg. 3 of Broderick et al. [20]. We focus on regimes where dozens of laps are feasible, which we demonstrate dramatically improves performance.

Affordable, but exact, batch optimization of $\mathcal{L}$ is possible by exploiting the additivity of statistics $M$, $S$. For each statistic we track a batch-specific quantity $M^b$, and a whole-dataset summary $M \triangleq \sum_{b=1}^{B} M^b$. After a local step at batch $b$ yields $\hat{s}^b, \hat{r}^b$, we update $M^b(\hat{s}^b)$ and $S^b(\hat{r}^b)$, *increment* each whole-dataset statistic by adding the new batch summary and subtracting the summary stored in memory from the previous visit, and store (or *memoize*) the new statistics for future iterations. This update cycle makes $M$ and $S$ consistent with the most recent assignments for *all* sequences. Memoization does require $\mathcal{O}(BK^2)$ more storage than SVI. However, this cost does not scale with the number of sequences $N$ or length $T$. Sparsity in transition counts $M$ may make storage cheaper.

At any point during memoized execution, we can evaluate $\mathcal{L}$ exactly for all data seen thus far. This is possible because nearly all terms in Eq. (6) are functions of only global parameters $\hat{\rho}, \hat{\omega}, \hat{\theta}, \hat{\tau}$ and sufficient statistics $M, S$. The one exception that requires local values $\hat{s}, \hat{r}$ is the entropy term $\mathcal{L}_{\text{entropy}}$. To compute it, we track a $(K + 1) \times K$ matrix $H^b$ at each batch $b$:

$$H_{0\ell}^b = -\sum_n \hat{r}_{n1\ell} \log \hat{r}_{n1\ell}, \quad H_{k\ell}^b = -\sum_n \sum_{t=1}^{T_n-1} \hat{s}_{ntk\ell} \log \frac{\hat{s}_{ntk\ell}}{\hat{r}_{ntk}}, \tag{12}$$

where the sums aggregate sequences $n$ that belong to batch $b$. Each entry of $H^b$ is non-negative, and given the whole-dataset entropy matrix $H = \sum_{b=1}^{B} H^b$, we have $\mathcal{L}_{\text{entropy}} = \sum_{k=0}^{K} \sum_{\ell=1}^{K} H_{k\ell}$.

## 4 State Space Adaptation via Birth, Merge, and Delete Proposals

Reliable nonparametric inference algorithms must quickly identify and create missing states. Split-merge samplers for HDP topic models [10, 11] are limited because proposals can only split an existing state into two new states, require expensive traversal of all data points to evaluate an acceptance ratio, and often have low acceptance rates [12]. Some variational methods for HDP topic models also dynamically create new topics [16, 21], but do not guarantee improvement of the global objective and can be unstable. We instead interleave stochastic birth proposals with delete and merge proposals, and use memoization to efficiently verify proposals via the exact full-dataset objective.

**Birth proposals.** Birth moves can create many new states at once while maintaining the monotonic increase of the whole-dataset objective, $\mathcal{L}$. Each proposal happens within the local step by trying to improve $q(z_n)$ for a single sequence $n$. Given current assignments $\hat{s}_n, \hat{r}_n$ with truncation $K$, the move proposes new assignments $\hat{s}'_n, \hat{r}'_n$ that include the $K$ existing states and some new states with index $k > K$. If $\mathcal{L}$ improves under the proposal, we accept and use the expanded set of states for all remaining updates in the current lap. To compute $\mathcal{L}$, we require candidate global parameters $\hat{\rho}', \hat{\omega}', \hat{\theta}', \hat{\tau}'$. These are found via a global step from candidate summaries $M', S'$, which combine

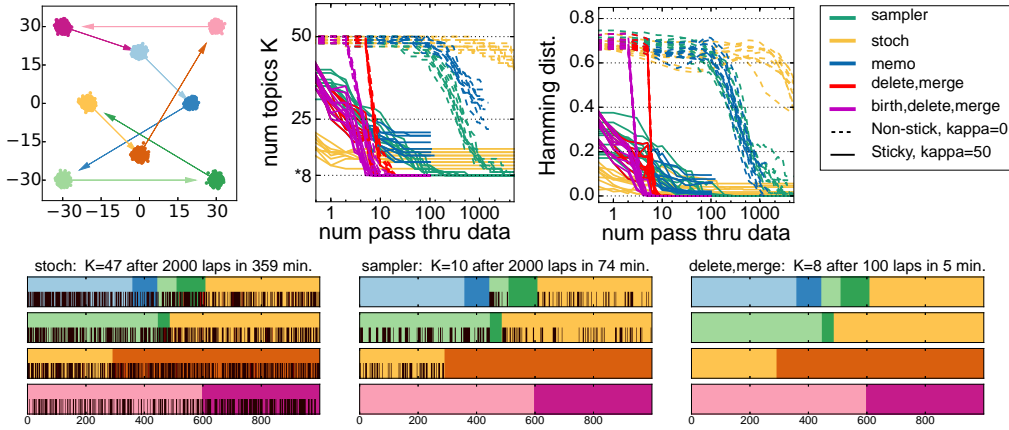

Figure 3: Toy data experiments (Sec. 5). *Top left:* Data sequences contain 2D points from 8 well-separated Gaussians with sticky transitions. *Top center:* Trace plots from initialization with 50 redundant states. Our state-adaptation algorithms (red/purple) reach ideal $K = 8$ states and zero Hamming distance regardless of whether a sticky (solid) or non-sticky (dashed) model is used. Competitors converge slower, especially in the non-sticky case because non-adaptive methods are more sensitive to hyperparameters. *Bottom:* Segmentations of 4 sequences by SVI, the Gibbs sampler, and our method under the non-sticky model ($\kappa = 0$). Top half shows true state assignments; bottom shows aligned estimated states. Competitors are polluted by extra states (black).

the new batch statistics $M'_b, S'_b$ and memoized statistics of other batches $M'_{\backslash b}, S'_{\backslash b}$ expanded by zeros for states $k > K$. See the supplement for details on handling multiple sequences within a batch.

The proposal for expanding $\hat{s}', \hat{r}'$ with new states can flexibly take any form, from very naïve to very data-driven. For data with "sticky" state persistence, we recommend randomly choosing one interval $[t, t + \delta]$ of the current sequence to reassign when creating $\hat{s}', \hat{r}'$, leaving other timesteps fixed. We split this interval into two contiguous blocks (one may be empty), each completely assigned to a new state. In the supplement, we detail a linear-time search that finds the cut point that maximizes the objective $\mathcal{L}_{\text{data}}$. Other proposals such as sub-cluster splits [11] could be easily incorporated in our variational algorithm, but we find this simple interval-based proposal to be fast and effective.

**Merge proposals.** Merge proposals try to find a less redundant but equally expressive model. Each proposal takes a pair of existing states $i < j$ and constructs a candidate model where data from state $j$ is reassigned to state $i$. Conceptually this reassignment gives a new value $\hat{s}'$, but instead statistics $M', S'$ can be directly computed and used in a global update for candidate parameters $\hat{\tau}', \hat{\rho}', \hat{\theta}'$.

$$S'_i = S_i + S_j, \quad M'_{:i} = M_{:i} + M_{:j}, \quad M'_{i:} = M_{i:} + M_{j:}, \quad M'_{ii} = M_{ii} + M_{jj} + M_{ji} + M_{ij}.$$

While most terms in $\mathcal{L}$ are linear functions of our cached sufficient statistics, the entropy $\mathcal{L}_{\text{entropy}}$ is not. Thus for each candidate merge pair $(i, j)$, we use $\mathcal{O}(K)$ storage and computation to track column $H'_{:i}$ and row $H'_{i:}$ of the corresponding merged entropy matrix $H'$. Because all terms in the $H'$ matrix of Eq. (12) are non-negative, we can lower-bound $\mathcal{L}_{\text{entropy}}$ by summing a subset of $H'$. As detailed in the supplement, this allows us to rigorously bound the objective $\mathcal{L}'$ for accepting multiple merges of distinct state pairs. Because many entries of $H'$ are near-zero, this bound is very tight, and in practice enables us to scalably merge many redundant state pairs in each lap through the data.

To identify candidate merge pairs $i, j$, we examine all pairs of states and keep those that satisfy $\mathcal{L}'_{\text{data}} + \mathcal{L}'_{\text{hdp-local}} + \mathcal{L}'_{\text{hdp-global}} > \mathcal{L}_{\text{data}} + \mathcal{L}_{\text{hdp-local}} + \mathcal{L}_{\text{hdp-global}}$. Because entropy must decrease after any merge ($\mathcal{L}'_{\text{entropy}} < \mathcal{L}_{\text{entropy}}$), this test is guaranteed to find all possibly useful merges. It is much more efficient than the heuristic correlation score used in prior work on HDP topic models [14].

**Deletes.** Our proposal to delete a rarely-used state $j$ begins by dropping row $j$ and column $j$ from $M$ to create $M'$, and dropping $S_j$ from $S$ to create $S'$. Using a *target dataset* of sequences with non-trivial mass on state $j$, $x' = \{x_n : \sum_{t=1}^{T_n} \hat{r}_{ntj} > 0.01\}$, we run global and local parameter updates to reassign observations from former state $j$ in a data-driven way. Rather than verifying on only the target dataset as in [14], we accept or reject the delete proposal via the whole-dataset bound $\mathcal{L}$. To control computation, we only propose deleting states used in 10 or fewer sequences.

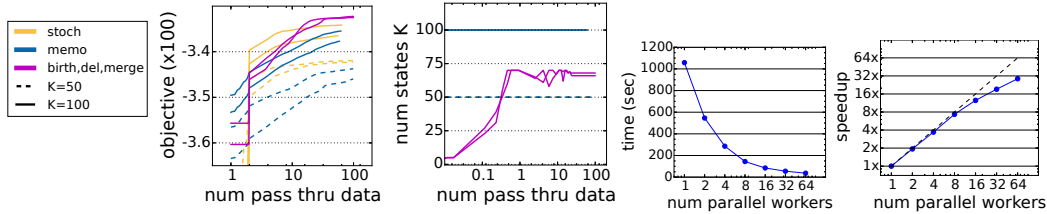

Figure 4: Segmentation of human epigenome: 15 million observations across 173 sequences (Sec. 5). *Left:* Adaptive runs started at 1 state grow to 70 states within one lap and reach better $\mathcal{L}$ scores than 100-state non-adaptive methods. Each run takes several days. *Right:* Wallclock times and speedup factors for a parallelized local step on 1/3 of this dataset. 64 workers complete a local step with $K = 50$ states in under one minute.

## 5    Experiments

We compare our proposed birth-merge-delete memoized algorithm to memoized with delete and merge moves only, and without any moves. We further run a blocked Gibbs sampler [6] that was previously shown to mix faster than slice samplers [22], and our own implementation of SVI for objective $\mathcal{L}$. These baselines maintain a fixed number of states $K$, though some states may have usage fall to zero. We start all fixed-$K$ methods (including the sampler) from matched initializations. See the supplement for futher discussion and all details needed to reproduce these experiments.

**Toy data.**    In Fig. 3, we study 32 toy data sequences generated from 8 Gaussian states with sticky transitions [8]. From an abundant initialization with 50 states, the sampler and non-adaptive variational methods require hundreds of laps to remove redundant states, especially under a non-sticky model ($\kappa = 0$). In contrast, our adaptive methods reach the ideal of zero Hamming distance within a few dozen laps regardless of stickiness, suggesting less sensitivity to hyperparameters.

**Speaker diarization.**    We study 21 unrelated audio recordings of meetings with an unknown number of speakers from the NIST 2007 speaker diarization challenge [23]. The sticky HDP-HMM previously achieved state-of-the-art diarization performance [6] using a sampler that required hours of computation. We ran methods from 10 matched initializations with 25 states and $\kappa = 100$, computing Hamming distance on non-speech segments as in the standard DER metric. Fig. 5 shows that within minutes, our algorithms consistently find segmentations better aligned to true speaker labels.

**Labelled $N = 6$ motion capture.**    Fox et al. [12] introduced a 6 sequence dataset with labels for 12 exercise types, illustrated in Fig. 1. Each sequence has 12 joint angles (wrist, knee, etc.) captured at 0.1 second intervals. Fig. 6 shows that non-adaptive methods struggle even when initialized abundantly with 30 (dashed lines) or 60 (solid) states, while our adaptive methods reach better values of the objective $\mathcal{L}$ and cleaner many-to-one alignment to true exercises.

**Large $N = 124$ motion capture.**    Next, we apply scalable methods to the 124 sequence dataset of [12]. We lack ground truth here, but Fig. 7 shows deletes and merges making consistent reductions from abundant initializations and births growing from $K = 1$. Fig. 7 also shows estimated segmentations for 10 representative sequences, along with skeleton illustrations for the 10 most-used states in this subset. These segmentations align well with held-out text descriptions.

**Chromatin segmentation.**    Finally, we study segmenting the human genome by the appearance patterns of regulatory proteins [24]. We observe 41 binary signals from [3] at 200bp intervals throughout a white blood cell line (CD4T). Each binary value indicates the presence or absence of an acetylation or methylation that controls gene expression. We divide the whole epigenome into 173 sequences (one per batch) with total size $T = 15.4$ million. Fig. 4 shows our method can grow from 1 state to 70 states and compete favorably with non-adaptive competitors. We also demonstrate that our parallelized local step leads to big 25x speedups in processing such large datasets.

## 6    Conclusion

Our new variational algorithms adapt HMM state spaces to find clean segmentations driven by Bayesian model selection. Relative to prior work [14], our contributions include a new bound for the sticky HDP-HMM, births with guaranteed improvement, local step parallelization, and better merge selection rules. Our multiprocessing-based Python code is targeted at genome-scale applications.

**Acknowledgments**    This research supported in part by NSF CAREER Award No. IIS-1349774. M. Hughes supported in part by an NSF Graduate Research Fellowship under Grant No. DGE0228243.

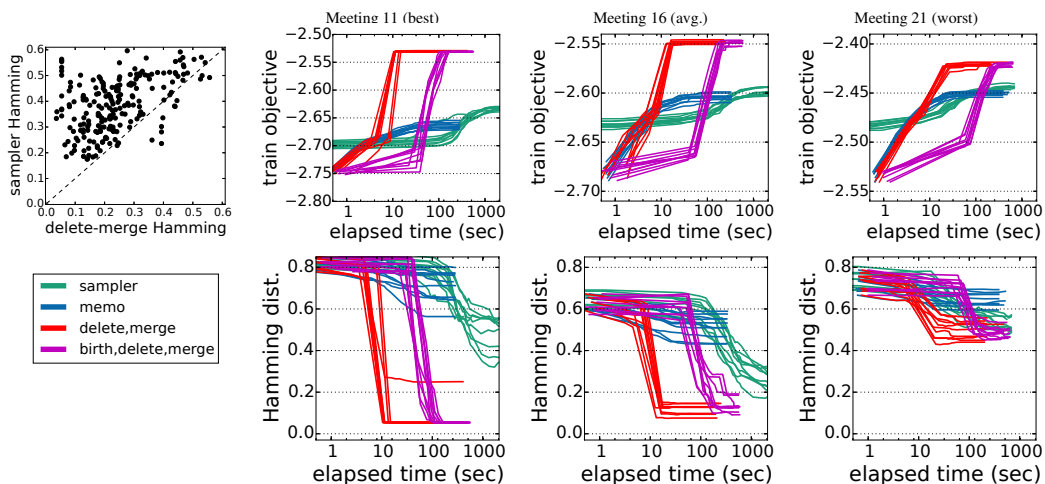

Figure 5: Method comparison on speaker diarization from common $K = 25$ initializations (Sec. 5). *Left:* Scatterplot of final Hamming distance for our adaptive method and the sampler. Across 21 meetings (each with 10 initializations shown as individual dots) our method finds segmentations closer to ground truth. *Right:* Traces of objective $\mathcal{L}$ and Hamming distance for meetings representative of good, average, and poor performance.

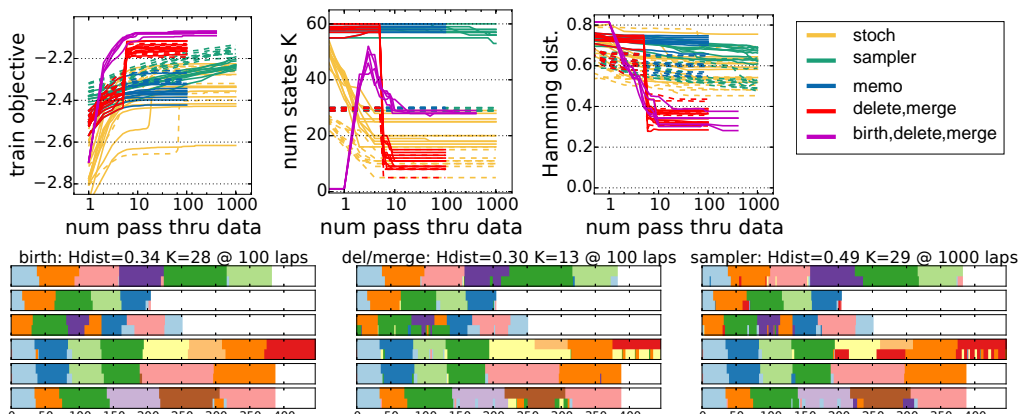

Figure 6: Comparison on 6 motion capture streams (Sec. 5). *Top:* Our adaptive methods reach better $\mathcal{L}$ values and lower distance from true exercise labels. *Bottom:* Segmentations from the best runs of birth/merge/delete (left), only deletes and merges from 30 initial states (middle), and the sampler (right). Each sequence shows true labels (top half) and estimates (bottom half) colored by the true state with highest overlap (many-to-one).

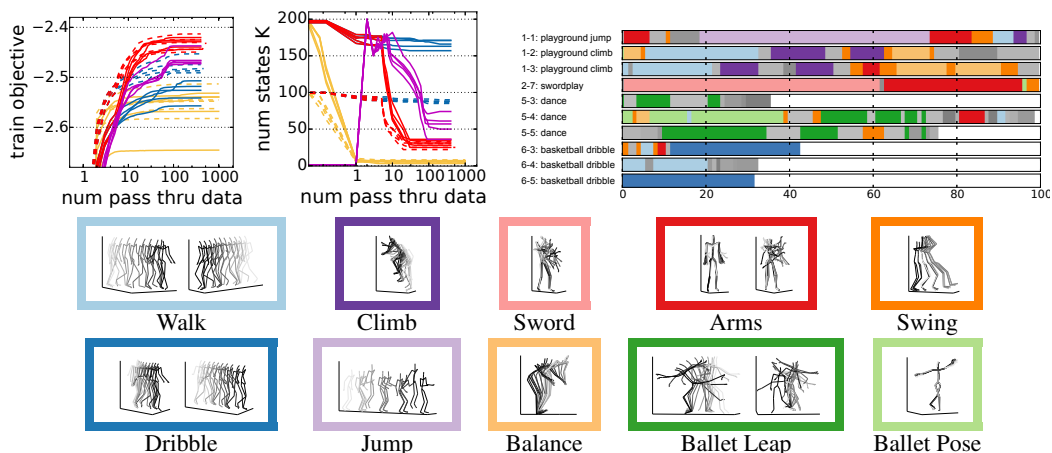

Figure 7: Study of 124 motion capture sequences (Sec. 5). *Top Left:* Objective $\mathcal{L}$ and state count $K$ as more data is seen. Solid lines have 200 initial states; dashed 100. *Top Right:* Final segmentation of 10 select sequences by our method, with id numbers and descriptions from `mocap.cs.cmu.edu`. The 10 most used states are shown in color, the rest with gray. *Bottom:* Time-lapse skeletons assigned to each highlighted state.

## Footnotes

[1] http://bitbucket.org/michaelchughes/x-hdphmm-nips2015/

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
