[Supplementary Material]

# Supplementary Material: Scalable Adaptation of State Complexity for Nonparametric Hidden Markov Models
## Paper published at NIPS 2015

**Michael C. Hughes**            MHUGHES@CS.BROWN.EDU
**William Stephenson**            WTSTEPHE@GMAIL.COM
**Erik B. Sudderth**            SUDDERTH@CS.BROWN.EDU
*Brown University Department of Computer Science, Providence, RI, USA.*

## Contents

## A. Experiment Details

Here, we describe how to recreate the experiments from Sec. 5 of the main paper. Our open-source Python package `bnpy`[1] contains learning algorithms for all variational methods (stochastic with fixed truncation, memoized with fixed truncation, and memoized with birth-delete-merge proposals). We have released a specialized repository called `x-hdphmm-nips2015`[2] that contains the exact scripts needed to call `bnpy` inference routines to reproduce our expeiments and plotting scripts to recreate our exact plots in Sec. 5 of the main paper. Thorough documentation can be found in the included README file.

**Model Hyperparameters.** The performance of our algorithms depends on several hyperparameters: stickiness $\kappa$, concentration parameters $\gamma, \alpha$, likelihood hyperparameters, etc. For each dataset, we document the values of hyperparameters used in each experiments both here in this supplement and in plain-text settings files within the `x-hdphmm-nips2015` codebase. See the mathematical formulas in Sec. C.1-C.2 to understand how these command line options translate into valid hyperparameter specifications.

**Gibbs sampler code.** Additionally, the `x-hdphmm-nips2015` project contains the Matlab code we adapted from Fox et al. (2011) to run our experiments. The three big changes to Fox's original code are listed here. First, we changed initialization to use the same starting segmentation as our variational algorithms. Second, we disabled hyperparameter resampling in favor of fixed values of $\gamma, \alpha, \kappa$ and other likelihod hyperparameters. Fixing these makes it possible to fairly compare to our variational methods, which all keep hyperparameters fixed. Finally, we adjusted how the Matlab code saves-to-disk so that it is consistent with corresponding runs in bnpy. Otherwise, the individual updates used in the sampler are the same as the original code release by Fox et al. (2011).

**Learning rate schedule.** For all datasets, we set the learning rate $\rho_t$ for stochastic variational at update iteration $t$ to $\rho_t = (1 + t)^{0.51}$. This is a fairly aggressive schedule, recommended in past work by **?**. Future work could tune this specifically for each dataset, but we chose to simplify the comparisons here. Note that under this setting, SVI reaches noticeably better objective scores than memoized methods on the chromatin experiments (Fig. 4 of the main paper), but performs worse on the larger-scale motion capture experiments (Fig. 7). See the file settings-bnpyHDPHMMstoch.txt for where this is defined in the codebase.

**Algorithm Hyperparameters.** The performance of our algorithms depends not only on some model parameters, but also on some inference parameters (setting batch sizes, number of laps to run, update schedules, number of merge moves to perform per lap, etc.). The exact settings used are all encoded as plain-text files in the `x-hdphmm-nips2015` codebase.

**Initialization.** Across all experiments in this paper, we used the same procedure to initialize algorithms given the provided data sequences and a specific number of clusters $K$. We call this procedure "random-contiguous-blocks", since it selects subwindows of data sequences at random and uses these to create the global likelihood parameters (via the standard global step). We specify it via the flag `--initname randcontigblocks` in bnpy. You can read more about this procedure in the included toy data IPython notebook.[3]

We found this worked well with all datasets here. To generalize to a new dataset, though, it is often advantageous to define the window length used as longer or shorter than our default, since some datasets have much longer segments than others. Also, higher-dimensional datasets can benefit from using more data in initialization. We thus suggest testing a few possible values of `--initBlockLen` for a new dataset to be sure of good performance. Of course, if the dataset has very little self-transition (lots of fast-switching states), another type of initialization may be preferred.

---

1. `http://bitbucket.org/michaelchughes/bnpy-dev`
2. `http://bitbucket.org/michaelchughes/x-hdphmm-nips2015`
3. `http://bitbucket.org/michaelchughes/x-hdphmm-nips2015/raw/master/notebooks/DDToyHMM-nipsexperiments.ipynb`

**Figure 1:** Algorithm comparison on toy dataset, using a non-sticky state transition model with $\kappa = 0$ (*top row*) and sticky model with $\kappa = 50$ (*middle row*). *Left column:* Objective function score $\mathcal{L}$ as more training data is seen. *Middle column:* Number of effective states $K$ used in the segmentation of the training dataset. For all algorithms, we count a state as present if it is assigned to at least one timestep of data in the estimated hard segmentation. *Right column:* Hamming distance between aligned segmentations and the ground truth segmentation. All non-birth algorithms are initialized using a common set of over-complete segmentations, with either $K = 50$ (dashed lines) or $K = 100$ states (solid lines). Comparing the non-sticky (top row) and sticky (middle row) models, we see that the sticky model encourages faster convergence for all algorithms. The non-sticky sampler takes thousands of laps through the dataset for Hamming distance to drop near zero, but it does reach that configuration, as illustrated in the segmentations in the bottom row. In contrast, the sticky sampler reaches this ideal by around 200 laps according to the middle trace plots. Both sticky and non-sticky models clearly prefer the ideal segmentation, but algorithm convergence is particularly sensitive to the sticky hyperparameter. Fixed-truncation methods (stochastic and memoized) are particularly vulnerable to slow convergence here. In contrast, our state-adaptive methods reach ideal configurations within 20 laps regardless of whether the model is sticky or not.

## A.1 Toy Data

**Dataset details.** Within the `x-hdphmm-nips2015` repository, this dataset is named `DDToyHMM`, which stands for diagonally-dominant toy HMM dataset. We have saved the exact dataset used for training as a MAT file within the datasets/ directory of this repo.

This toy dataset has $N = 32$ sequences divided into $B = 8$ batches. Each sequence has length $T_n = 1000$. When we show segmentations in the the main paper, we always show segmentations for sequences 1, 3, 5, and 7, which together form a good representation of all 8 true states.

Each observation is a 2D real vector $x_{nt} \in \mathbb{R}^2$. We use a full-covariance Gaussian for likelihood $F$ and a corresponding Wishart distribution for the prior $H$.

**Model hyperparameters.**

```
--gamma 10
--alpha 0.5
--startAlpha 5
--stickyKappa 0 or 50
--nu D+2
--ECovMat eye
--sF 1.0
--kappa 1e-7
--MMat zero
```

**Detailed experiment.** In Fig. 1, we show trace plots that compare the progress of various algorithms under two settings: non-sticky dynamics ($\kappa = 0$) and sticky dynamics ($\kappa = 50$). Comparing non-sticky and sticky models, we see that the sticky model generally encourages faster convergence for all algorithms. In particular, for the Gibbs sampler of Fox et al. (2011), the non-sticky sampler takes thousands of laps through the dataset for Hamming distance to drop near zero, but it does reach that configuration, as illustrated by the segmentations in the bottom row. In contrast, the sticky sampler reaches this ideal by around 200 laps according to the trace plots. Thus, the performance of the sampler can be quite sensitive to the value of the provided sticky hyperparameter. We thank an anonymous reviewer for suggesting this detailed analysis.

Note that across Fig. 1 in both sticky and non-sticky cases, our adaptive algorithms with birth/merge/delete proposals eliminate the redundant states more quickly than non-adaptive competitors. Our proposal moves enable fast convergence regardless of the hyperparameter values, suggested that algorithms with greater power to escape local optima can avoid some sensitivity exhibited by more limited methods.

## A.2 Speaker Diarization

Within the `x-hdphmm-nips2015` repository, this dataset is named `SpeakerDiar`. Our experiment repository saves the exact dataset used (including relevant preprocessing, which we duplicated from Fox et al. (2011)), in a MAT file under the datasets/ directory.

There are $N = 21$ sequences, which have no overlap in terms of common speakers. Thus, we process each one independently. This makes "memoized" inference equivalent to full-dataset inference because there is only one batch. It also makes stochastic inference irrelevant, so we only compare to the sampler as a baseline method.

We use a full covariance Gaussian likelihood $F$ with corresponding Wishart prior. The relevant hyperparameters are:

```
--gamma 10
--alpha 0.5
--startAlpha 10
--stickyKappa 100
--nu D+2
--ECovMat covdata
--sF 0.5
--kappa 1e-7
--MMat zero
```

For Hamming distance computations on this dataset, we utilize the provided annotations of each sequence into "background" (non-speech) and "foreground" (speech) states. We only count

timesteps labeled as foreground in the distance computation, and ignore any assignments to timesteps labelled background. Our dataset clearly marks background labels with negative integer labels, while foreground states have non-negative labels $\{0, 1, ...\}$.

### A.3 Motion capture dataset.

Within the `x-hdphmm-nips2015` repository, we have the smaller $N = 6$ dataset named `MoCap6`, and the larger one `MoCap124`. The repository saves the exact dataset used (including relevant preprocessing, which we duplicated from Fox et al. (2014)), in a MAT file under the datasets/ directory.

We use a first order AR Gaussian likelihood. We process each of the $N = 6$ sequences as its one batch. For the larger $N = 124$ dataset, we divide sequences into 20 batches.

```
--gamma 10
--alpha 0.5
--startAlpha 10
--stickyKappa 100
--nu D+2
--ECovMat diagcovfirstdiff
--sF 0.5
--VMat same
--sV 0.5
--MMat eye
```

Scripts for visualizing the skeleton trace of a specific data segment can be found in a specialized git repository [4].

**Discussion of Fig. 6 of main paper:** For non-adaptive methods on the 6-sequence dataset, we compare each algorithm initialized from abundant initializations of 30 (dashed) and 60 (solid) states. It seems the 30 state models are slightly preferred (especially for the sampler). However, for our adaptive models with deletes and merges (red curves) and with births (purple), the number of states in the initialization does not seem to matter too much.

**Discussion of Fig. 7 of main paper:** For non-adaptive methods on the 124-sequence dataset, we compare each algorithm initialized from abundant initializations of 100 (dashed) and 200 (solid) states. For the stochastic method (SVI, yellow curves), under all initializations we see a rapid drop in the number of states used during the first lap. To explain this, remember that with 20 batches for 124 sequences, each batch will have around 6 sequences. From the segmentation figure, it is clear that state usage patterns have lots of variety across sequences, which each sequence only using a handful of states. The aggressive learning rate we use in the first lap will tend to severely downweight any initial states not used in the first few batches, which explains the rapid drop in Fig. 7. In contrast, the memoized method (blue) is designed to use global information for each parameter update, not just the current batch. We further enforce this by delaying the first global update until at least 50 sequences are seen. This makes the memoized results a large improvement on the stochastic results for this dataset.

Among non-adaptive memoized runs, we see a clear preference in the trace plots for 100 states over 200 states (dashed lines reach higher objective scores). Furthermore, using delete and merge moves only (red) shows that we can reduce down to about 30 states and reach even higher levels of performance. Similarly, starting from 1 state with birth moves (purple), we can grow to nearly comparable levels of performance. We hope to answer why the purple curves do not quite reach the performance of the red curves in future work. Regardless, the set of adaptive methods reach high objective scores much more consistently than non-adaptive methods.

---

4. http://github.com/michaelchughes/mocap6dataset/

### A.4 Chromatin epigenomic dataset

We used the binary data for chromatin marker protein presence and absence for the whole genome preprocessed and made available by Ernst and Kellis (2010). We divided up the very long original 24 sequences (one per chromosome) into smaller sets to test the ability of our algorithms to handle many batches. To divide each sequence, we searched for intervals with at least 50 consecutive all-zero observations, which are somewhat common since much of the genome is "junk". We picked division points in the middle of these empty segments to use to split up into more manageable size sequences, while avoiding artifacts at the starts of each sequence as much as possible.

In the end, we obtained $N = 173$ sequences, ranging in size from $T_n = 10000$ to $T_n = 200,000$ timesteps (aka observations). Each observation is a 41-dimensional binary vector. For the likelihood $F$, we used a Bernoulli with a corresponding Beta prior. The relevant hyperparameters are:

```
--gamma 10
--alpha 0.5
--startAlpha 10
--stickyKappa 100
--lam1 0.1
--lam0 0.3
```

which means the Beta prior was equal to: $\phi_{kd} \sim \text{Beta}(0.1, 0.3)$.

## B. Variational objective function $\mathcal{L}$

Here, we give complete, final expressions for computing each term of our objective $\mathcal{L}$ in terms of the global free parameters and local free parameters. Where possible, we use summary statistics $M(\hat{s}), S(x, \hat{r})$ instead of the local parameters $\hat{s}, \hat{r}$ so it is clear how this computation works in the memoized setting. First, we recall our definition of $\mathcal{L}$ as a composition of additive terms from the main paper:

$$\mathcal{L} = \mathcal{L}_{\text{hdp-global}} + \mathcal{L}_{\text{hdp-local}} + \mathcal{L}_{\text{data}} + \mathcal{L}_{\text{entropy}} \tag{1}$$

### B.1 Term $\mathcal{L}_{\text{hdp-global}}$

$$
\begin{aligned}
\mathcal{L}_{\text{hdp-global}}(\hat{\rho}, \hat{\omega}) &\triangleq \mathbb{E}_q \left[ \sum_{k=1}^{K} \log \frac{p(u_k \mid \gamma)}{q(u_k \mid \hat{\rho}_k, \hat{\omega}_k)} \right] \\
&= \sum_{k=1}^{K} c_B(1, \gamma) - c_B(\hat{\rho}_k \hat{\omega}_k, (1 - \hat{\rho}_k)\hat{\omega}_k) \\
&\quad + \left(1 - \hat{\rho}_k \hat{\omega}_k\right) \mathbb{E}_q[\log u_k] + \left(\gamma - (1 - \hat{\rho}_k)\hat{\omega}_k\right) \mathbb{E}_q[\log 1 - u_k]
\end{aligned}
\tag{2}
$$

where we have $\mathbb{E}_q[\log u_k] = \psi(\hat{\rho}_k \hat{\omega}_k) - \psi(\omega_k)$, and $\mathbb{E}_q[\log 1 - u_k] = \psi((1 - \hat{\rho}_k)\hat{\omega}_k) - \psi(\omega_k)$.

### B.2 Term $\mathcal{L}_{\text{hdp-local}}$

We can write this term in two parts: $\mathcal{L}_{\text{hdp-local}} = \mathcal{L}_{\text{sur}} + \mathcal{L}_{\text{hdplocaltrans}}$.

**Surrogate term that bounds the Dirichlet cumulant function.** This term requires the lower bound developed in Sec. D.2

$$\mathcal{L}_{\mathrm{sur}}(\hat{\rho}, \hat{\omega}) = K \log \alpha_0 + K^2 \log \alpha + K \big( \log(\kappa) - \log(\alpha + \kappa) \big) \tag{3}$$
$$+ (\log(\alpha + \kappa) - \log(\kappa)) \sum_{k=1}^{K} \mathbb{E}_q[\beta_k]$$
$$+ \sum_{k=1}^{K} K \mathbb{E}_q[\log u_k] + [K(K+1-k)+1]\mathbb{E}_q[\log 1 - u_k]$$

where the expectations $\mathbb{E}_q[\log u_k]$ and $\mathbb{E}_q[\log 1 - u_k]$ are given above, and for each $k = \{1, 2, \dots K\}$ we have:

$$\mathbb{E}_q[\beta_k] = \hat{\rho}_k \prod_{m=1}^{k-1}(1 - \hat{\rho}_m) \tag{4}$$
$$\mathbb{E}_q[\beta_{>K}] = \prod_{m=1}^{K}(1 - \hat{\rho}_m).$$

**Remainder term that gathers transition factors.** Here, we gather all terms related to the transition factors $q(\pi)$ and generation of state sequence $z$.

$$\mathcal{L}_{\mathrm{hdplocaltrans}}(\hat{s}, \hat{\rho}, \hat{\theta}) \triangleq \mathbb{E}_q\left[ \log p(z|\pi) + \sum_{k=0}^{K} \log \frac{p(\pi_k | \alpha_k \beta + \kappa \delta(k))}{q(\pi_k | \hat{\theta}_k)} \right] - \mathcal{L}_{\mathrm{sur}} \tag{5}$$
$$= -\sum_{k=0}^{K} c_D(\hat{\theta}_k) + \sum_{k=0}^{K} \sum_{\ell=1}^{K+1} \left( M_{k\ell}(\hat{s}) + \alpha_k \mathbb{E}_q[\beta_\ell] + \kappa \delta_\ell(k) - \hat{\theta}_{k\ell} \right) P_{k\ell}(\hat{\theta})$$

Where we have summary statistics

$$P_{k\ell}(\hat{\theta}) \triangleq \mathbb{E}_q[\log \pi_{k\ell}] = \psi(\hat{\theta}_{k\ell}) - \psi(\textstyle\sum_{m=1}^{K+1} \hat{\theta}_{km}), \qquad k \in \{0, 1, \dots K\} \tag{6}$$
$$M_{k\ell}(\hat{s}) \triangleq \begin{cases} \mathbb{E}_q[\sum_n \sum_{t=1}^{T_n - 1} \delta_k(z_{nt})\delta_\ell(z_{nt+1})] = \sum_{n=1}^{N} \sum_{t=1}^{T-1} \hat{s}_{ntk\ell} & k \in \{1, 2, \dots K\} \\ \mathbb{E}_q[\sum_{n=1}^{N} \delta_\ell(z_{n1})] = \sum_{n=1}^{N} \hat{r}_{n1\ell} & k = 0 \end{cases}$$

Where the row of $M$ with index 0 corresponds to a special starting state.

## B.3 Term $\mathcal{L}_{\mathbf{data}}$

Gathering all terms relevant to data generation, we have

$$\mathcal{L}_{\mathrm{data}}(x, \hat{\tau}, \hat{r}) \triangleq \mathbb{E}_q\left[ \log p(x|z, \phi) + \log \tfrac{p(\phi)}{q(\phi)} \right] \tag{7}$$
$$= \sum_{k=1}^{K} c_H(\bar{\tau}) - c_H(\hat{\tau}_k) + (S_k + \bar{\tau} - \hat{\tau}_k)^T \mathbb{E}_{q(\phi_k | \hat{\tau}_k)}[\phi_k \quad c_F(\phi_k)]$$

The expectations of $\phi_k$ and $c_F(\phi_k)$ depend on the chosen densities for likelihood $F$ and prior $H$, but generally work out to closed-form functions of the parameters $\hat{\tau}_k$ and $\bar{\tau}$.

## B.4 Term $\mathcal{L}_{\mathbf{entropy}}$

The final term gives the entropy of the assignment distributions.

$$\mathcal{L}_{\mathrm{entropy}}(\hat{s}, \hat{r}) \triangleq -\sum_{n=1}^{N} \mathbb{E}_q\left[ \log q(z_n) \right], \qquad \mathbb{E}_q\left[ \log q(z_n) \right] = \hat{r}_{n1k} \log \hat{r}_{n1k} + \sum_{t=1}^{T_n - 1} \hat{s}_{ntk\ell} \log \frac{\hat{s}_{ntk\ell}}{\hat{r}_{ntk}}. \tag{8}$$

which can also be computed from the entropy summary quantities in matrix $H$, as described in the main paper.

$$\mathcal{L}_{\mathrm{entropy}}(H) = \sum_{b=1}^{B} \sum_{k=0}^{K} \sum_{\ell=1}^{K} H_{k\ell}^{b} \tag{9}$$
$$H_{0\ell}^{b} = -\sum_n \hat{r}_{n1\ell} \log \hat{r}_{n1\ell}, \quad H_{k\ell}^{b} = -\sum_n \sum_{t=1}^{T_n - 1} \hat{s}_{ntk\ell} \log \frac{\hat{s}_{ntk\ell}}{\hat{r}_{ntk}}, \tag{10}$$

## C. Data-generation densities $F$ and $H$

### C.1 $F$: Multivariate Gaussian, $H$: Gaussian-Wishart

**Multivariate Gaussian**  When the likelihood function $F$ is Gaussian, each state $k$ has two parameters:

$\Lambda_k$  $D \times D$ symmetric, pos. definite matrix  *precision matrix*

$\mu_k$  real vector, size $D$  *mean vector*

Where we have the useful expectations:

$$\mathbb{E}_F[x_{nt}] = \mu_k \qquad \mathbb{E}_F[(x_{nt} - \mu_k)(x_{nt} - \mu_k)^T] = \Lambda_k^{-1} \tag{11}$$

The likelihood density $F$ is defined as:

$$F: \qquad \log p(x_{nt}|\mu_k, \Lambda_k) = -\frac{D}{2}\log(2\pi) + (\frac{1}{2})\log|\Lambda_k| \tag{12}$$
$$- \frac{1}{2}\text{vec}(x_n x_n^T)^T \text{vec}(\Lambda_k) + x_n^T \Lambda_k \mu_k - \frac{1}{2}\mu^T \Lambda_k \mu_k$$

In exponential family notation, we have sufficient statistic:

$$s_F(x_{nt}) = \begin{bmatrix} 1 & x_n & \text{vec}(x_n x_n^T) \end{bmatrix} \tag{13}$$

**Multivariate Gaussian-Wishart**  The conjugate prior $H$ is a Gaussian-Wishart distribution. It has four parameters:

$\nu$  $\nu > 0$  *degrees-of-freedom, larger means more unimodal*

$B$  $D \times D$ symmetric, positive definite matrix  *scale matrix*

$\kappa$  $\kappa > 0$  *controls precision on $\mu$ relative to $\Lambda$*

$m$  real vector, size $D$  *mean of $\mu$*

We generally use $m = 0$, although we give the next two equations with general $m$ for the sake of completeness. Under this prior, we have the following useful expectations:

$$\mathbb{E}_H[\mu_k] = m, \qquad \mathbb{E}_H[\Lambda_k^{-1}] = \frac{B}{\nu - D - 1} \tag{14}$$

The exact density of $H$ is defined as:

$$H: \qquad \log p(\mu_k, \Lambda_k|\bar{\tau}) = \log p(\mu_k|\bar{\tau}, \Lambda_k) + \log p(\Lambda_k|\bar{\tau}) \tag{15}$$
$$= c(\nu, B, m, \kappa) + (\frac{v - D}{2})\log|\Lambda| - \frac{1}{2}\text{vec}(B + \kappa mm^T)^T \text{vec}(\Lambda)$$
$$+ (\kappa m)^T \Lambda \mu - \frac{1}{2}(\kappa)\mu^T \Lambda \mu$$

where we have the cumulant function for $H$ defined as:

$$c_H(\nu, B, m, \kappa) = -\frac{D}{2}\log(2\pi) - \frac{D(D-1)}{4}\log\pi - \frac{D\nu}{2}\log 2 \tag{16}$$
$$- \sum_{d=1}^{D} \log\Gamma\left(\frac{\nu + 1 - d}{2}\right) + \frac{D}{2}\log\kappa + \frac{\nu}{2}\log|B|$$

**Hyperparameters Options for Gaussian-Wishart Priors** Within `bnpy`, several command line options specify the hyperparameters $\nu, B, \kappa, m$ in our experiments:

```
--ECovMat [eye,covdata,diagcovfirstdiff]
--sF <scalar>
--kappa <scalar>
--nu <scalar>
```

Given these arguments, variables `kappa` and `sF` are straightforwardly set to provided values. The degrees-of-freedom variable is set so that $\nu = \min(\nu, D + 2)$, which guarantees the prior has a valid mean.

Finally, the `ECovMat` parameter describes how to set $B$ by constraining the *mean* or expectation of the covariance matrix $\Sigma = \mathbb{E}[\Lambda_k^{-1}]$ under the Wishart prior.

$$\Sigma = s_F \cdot \begin{cases} I_D & \text{ECovMat = 'eye'} \\ \sum_{n=1}^{N} \sum_{t=1}^{T_n} (x_{nt} - \bar{x})(x_{nt} - \bar{x})^T & \text{ECovMat = 'covdata'} \\ \sum_{n=1}^{N} \sum_{t=1}^{T_n} (y_{nt} - \bar{y})(y_{nt} - \bar{y})^T & \text{ECovMat = 'covfirstdiff'} \end{cases} \tag{17}$$

where $y_{nt} = x_{nt} - x_{nt-1}$ is the *first difference* at observation index $(n, t)$.

Thus, together the input keyword options `sF` and `ECovMat` determine the value of $\Sigma$, the expected covariance matrix. We can then easily set the parameter $B$ accordingly: $B = (\nu - D - 1)\Sigma$.

## C.2 $F$: Multivariate Auto-Regressive, $H$: Matrix Normal-Wishart

When the likelihood function $F$ is first-order auto-regressive Gaussian [5], each state $k$ has two parameters:

$$\begin{array}{lll} \Lambda_k & D \times D \text{ symmetric positive definite} & \textit{precision matrix} \\ A_k & D \times D \text{ matrix} & \textit{regression coefficients} \end{array} \tag{18}$$

where parameter $A_k$ defines the expected value of each successive data item: $\mathbb{E}_F[x_{nt}|x_{nt-1}] = A_k x_{nt-1}.$, while parameter $\Lambda_k$ defines the covariance matrix of state $k$.

The likelihood density $F$ is defined as:

$$F: \qquad \log p(x_{nt}|A_k, \Lambda_k, x_{nt-1}) = \log \text{Normal}(x_{nt}|A_k x_{nt-1}, \Lambda_k^{-1}) \tag{19}$$

$$= -\frac{D}{2}\log(2\pi) + \frac{1}{2}\log|\Lambda_k| - \frac{1}{2}(x_{nt} - A_k x_{nt-1}^T)\Lambda(x_{nt} - A_k x_{nt-1})$$

Expanding the quadratic form, we can rewrite it as several trace products.

$$(x_{nt} - A_k x_{nt-1})^T \Lambda_k (x_{nt} - A_k x_{nt-1}) = \text{tr}(\Lambda_k x_{nt} x_{nt}^T) - 2\text{tr}(\Lambda_k A_k x_{nt-1} x_{nt}^T) + \text{tr}(A_k^T \Lambda_k A_k x_{nt-1} x_{nt-1}^T) \tag{20}$$

In exponential family notation, we have sufficient statistic:

$$s_F(x_{nt}, x_{nt-1}) = \begin{bmatrix} 1 & \text{vec}(x_{nt} x_{nt}^T) & \text{vec}(x_{nt-1} x_{nt}^T) & \text{vec}(x_{nt-1} x_{nt-1}^T) \end{bmatrix} \tag{21}$$

**Matrix-Normal Wishart** The conjugate prior $H$ when $F$ is an AR process is the Matrix-Normal Wishart distribution. It has four parameters:

$$\begin{array}{lll} \nu & \nu > 0 & \textit{degrees-of-freedom, larger means more unimodal} \\ B & D \times D \text{ symmetric, positive definite matrix} & \textit{scale matrix for} \Lambda_k \\ V & D \times D \text{ symmetric, positive definite matrix} & \textit{determines covariance of } A_k \text{ relative to } \Lambda_k \\ M & D \times D & \textit{mean of } A_k \end{array}$$

---

5. Generalization to $R$-order dynamics is possible, but we do first-order only for simplicity.

We have the useful expectations:

$$\mathbb{E}[\Lambda_k^{-1}] = \frac{1}{\nu - D + 1} B \tag{22}$$

$$\mathbb{E}[A_k] = M$$

$$\text{Cov}[A_k] = \frac{1}{\nu - D + 1}(V^{-1} \times B)$$

The log density of $H$ is defined as

$$\log \text{MatrixNormalWish}(A_k, \Lambda_k | \nu, B, M, V^{-1}) = c_{MNW}(\nu, B, M, V) \tag{23}$$

$$+ \frac{\nu - 1}{2} \log |\Lambda_k|$$

$$- \frac{1}{2} \text{tr}\left([A_k^T \Lambda_k A_k] V\right)$$

$$+ \text{tr}\left([\Lambda_k A_k] V M^T\right)$$

$$- \frac{1}{2} \text{tr}\left([\Lambda_k](B + M V M^T)\right)$$

The cumulant function $c_{MNW}$ of the Matrix Normal Wishart is given by:

$$c_{MNW}(\nu, B, M, V) = c_{Wish}(\nu, B) + c_{MN}(M, V) \tag{24}$$

$$c_{Wish}(\nu, B) = -\frac{D(D-1)}{4} \log \pi - \frac{D}{2}(\log 2)\nu - \sum_{d=1}^{D} \log \Gamma\left(\frac{\nu + 1 - d}{2}\right) + \frac{\nu}{2} \log |B|$$

$$c_{MN}(M, V) = -\frac{D^2}{2} \log 2\pi + \frac{D}{2} \log |V|$$

**Hyperparameters Options for Matrix-Normal-Wishart Priors** We specify several command line options to specify the hyperparameters $\nu, B, V, M$ in our experiments.

```
--nu <scalar>
--ECovMat [eye,covdata,diagcovfirstdiff]
--sF <scalar>
--VMat [eye, same]
--sV <scalar>
--MMat [eye, zero]
```

Given these keywords, we transform `ECovMat` and `sF` options into a numerical value for $B$ via the same procedure we use for Gaussians, in Eq. (17). Similarly, we enforce $\nu = \min(\nu, D + 2)$ so that the prior specifies a valid mean.

The `VMat` option works as follows:

$$V = s_V \cdot \begin{cases} I_D & \text{eye} \\ \Sigma^{-1} & \text{same} \end{cases} \tag{25}$$

where $\Sigma$ is the matrix defined in Eq. (17). Specifying 'same' for the value of VMat results in an identity matrix covariance for parameter $A_k$.

The `MMat` option works as follows:

$$M = s_M \cdot \begin{cases} I_D & \text{eye} \\ 0_D & \text{zero} \end{cases} \tag{26}$$

where $I_D$ and $0_D$ are the identity matrix and the matrix of all zeros, respectively, each of size $D \times D$.

## D. Surrogate bound derivation details

### D.1 Bound on cumulant function of Dirichlet

As in the main paper, we define the cumulant function $c_D$ of the Dirichlet distribution as

$$c_D(\alpha\beta) = c_D(\alpha\beta_1, \alpha\beta_2, \dots \alpha\beta_K, \alpha\beta_{K+1}) \triangleq \log\Gamma(\alpha) - \sum_{k=1}^{K+1} \log\Gamma(\alpha\beta_k) \qquad (27)$$

where $\alpha > 0$ is a positive scalar, and $\beta = \{\beta_k\}_{k=1}^{K+1}$ is a vector of positive numbers that sum-to-one. The log-Gamma function $\log\Gamma(\cdot)$ has the following series representation[6] for scalar input $x > 0$:

$$-\log\Gamma(x) = \log x + \gamma x + \sum_{n=1}^{\infty} \left(\log\left(1 + \frac{x}{n}\right) - \frac{x}{n}\right) \qquad (28)$$

where $\gamma \approx .57721$ is the Euler-Mascheroni constant.

Substituting this expansion for every $\log\Gamma(\cdot)$ in the definition of $c_D$, we find

$$c_D(\alpha\beta) = -\log\alpha - \gamma\alpha - \sum_{n=1}^{\infty}\left(\log\left(1 + \frac{\alpha}{n}\right) - \frac{\alpha}{n}\right) \qquad (29)$$

$$+ \sum_{k=1}^{K+1}\left[\log\alpha\beta_k + \gamma\alpha\beta_k + \sum_{n=1}^{\infty}\left(\log\left(1 + \frac{\alpha\beta_k}{n}\right) - \frac{\alpha\beta_k}{n}\right)\right]$$

Here, all the infinite sums are convergent. This allows some regrouping, and we find that several terms cancel to zero. Our expression for $c_D(\alpha\beta)$ now simplifies to:

$$c_D(\alpha\beta) = -\log\alpha + \sum_{k=1}^{K+1}\log\alpha\beta_k \qquad (30)$$

$$+ \sum_{n=1}^{\infty}\left(\log\left(\prod_{k=1}^{K+1}\left(1 + \frac{\alpha\beta_k}{n}\right)\right) - \log\left(1 + \frac{\alpha}{n}\right)\right)$$

Finally, via the binomial product expansion below, we realize that the infinite sum must be larger than zero.

$$\prod_{k=1}^{K+1}\left(1 + \frac{\alpha\beta_k}{n}\right) = 1 + \sum_{k=1}^{K+1}\frac{\alpha\beta_k}{n} + \text{pos. const.} \quad \rightarrow \quad \prod_{k=1}^{K+1}\left(1 + \frac{\alpha\beta_k}{n}\right) \geq \left(1 + \frac{\alpha}{n}\right) \qquad (31)$$

Thus, by simply leaving off the infinite sum from Eq. (31) we have a valid lower bound on $c_D(\cdot)$:

$$c_D(\alpha\beta) \geq -\log\alpha + \sum_{k=1}^{K+1}\log\alpha\beta_k \qquad (32)$$

Expanding $\log\alpha\beta_k = \log\alpha + \log\beta_k$, we can further simplify to

$$c_D(\alpha\beta) \geq c_{sur}(\alpha, \beta) \triangleq K\log\alpha + \sum_{k=1}^{K+1}\log\beta_k \qquad (33)$$

---

6. http://mathworld.wolfram.com/LogGammaFunction.html

## D.2 Bound on cumulant function of sticky Dirichlet.

Applying the above bound to the case of the sticky hyperparameter gives an equation analogous to Eq. (33):

$$c_D(\alpha\beta_k + \delta_k\kappa) \geq K\log\alpha - \log(\alpha + \kappa) + \log(\alpha\beta_k + \kappa) + \sum_{\substack{m=1 \\ m\neq k}}^{K+1} \log(\beta_m) \tag{34}$$

Evaluating this term requires computing $\mathbb{E}_q[\log(\alpha\beta_k + \kappa)]$, which has no closed form in terms of elementary functions. Instead of calculating this directly, we use the concavity of logarithms to lower bound this term:

$$\log(\alpha\beta_k + \kappa) \geq \beta_k \log(\alpha + \kappa) + (1 - \beta_k)\log(\kappa) \tag{35}$$

$$= \beta_k\Big(\log(\alpha + \kappa) - \log(\kappa)\Big) + \log\kappa$$

We justify this bound by noting that for any practical value of $\kappa$ (say, $\kappa \sim 100$), this inequality is very tight, as shown in Fig. 2 in the main paper. Empirically, we find that $\kappa$ almost always needs to be either zero, in which case we do not apply the bound, or in the low hundreds, in which case the gap in the bound is completely negligible.

Plugging this bound in Eq. (34), we find

$$c_D(\alpha\beta_k + \delta_k\kappa) \geq c_{sur-\kappa}(\alpha, \kappa, \beta, k) \triangleq K\log\alpha + \log(\kappa) - \log(\alpha + \kappa) + \sum_{\substack{m=1 \\ m\neq k}}^{K+1} \log(\beta_m) \tag{36}$$

$$+ \beta_k\Big(\log(\alpha + \kappa) - \log(\kappa)\Big)$$

This equation gives us a surrogate bound on the cumulant function for a single sticky transition vector. We next need to compute the sum of $K$ sticky cumulant functions, plus one non-sticky cumulant function for the starting state.

Using our surrogate functions, we have

$$c_{sur}(\alpha, \beta) + \sum_{k=1}^{K} c_{sur-\kappa}(\alpha, \kappa, \beta, k) = K\log\alpha_0 + K^2\log\alpha \tag{37}$$

$$+ K\big(\log(\kappa) - \log(\alpha + \kappa)\big)$$

$$+ \Big(\log(\alpha + \kappa) - \log(\kappa)\Big)\sum_{k=1}^{K}\beta_k$$

$$+ \sum_{k=1}^{K+1}\log\beta_k + \sum_{k=1}^{K}\sum_{\substack{m=1 \\ m\neq k}}^{K+1}\log(\beta_m)$$

In the last line, the first sum comes from the stating state's cumulant function, the second nested sum comes from the others. We can combine these two terms to find that

$$c_{sur}(\alpha, \beta) + \sum_{k=1}^{K} c_{sur-\kappa}(\alpha, \kappa, \beta, k) = K\log\alpha_0 + K^2\log\alpha + K\big(\log(\kappa) - \log(\alpha + \kappa)\big) \tag{38}$$

$$+ \Big(\log(\alpha + \kappa) - \log(\kappa)\Big)\sum_{k=1}^{K}\beta_k$$

$$+ \log\beta_{K+1} + K\sum_{k=1}^{K+1}\log(\beta_k)$$

Finally, we can rewrite these sums of surrogate cumulants in terms of variable $u$ instead of $\beta$, since the transformation is deterministic. We find

$$c_{sur}(\alpha, \beta(u)) + \sum_{k=1}^{K} c_{sur-\kappa}(\alpha, \kappa, \beta(u), k) = K \log \alpha_0 + K^2 \log \alpha + K\big(\log(\kappa) - \log(\alpha + \kappa)\big) \quad (39)$$

$$+ \Big( \log(\alpha + \kappa) - \log(\kappa) \Big) \sum_{k=1}^{K} \beta_k(u)$$

$$+ \sum_{k=1}^{K} \Big( K \log u_k + [K(K+1-k)+1]\log(1-u_k) \Big)$$

We can now easily compute expectations of Eq. (39), since $\mathbb{E}_q[\beta_k]$ and $\mathbb{E}_q[\log u_k]$ have known closed forms when $q(u_k) = \text{Beta}(\hat{\rho}_k \hat{\omega}_k, (1 - \hat{\rho}_k)\hat{\omega}_k)$.

## E. Global update for $q(u|\hat{\rho}, \hat{\omega})$

Here, we derive the results needed to perform numerical optimization of the top-level beta parameters, $\hat{\rho}$ and $\hat{\omega}$. We only show the results with the sticky hyperparameter; optimization for the case of $\kappa = 0$ is covered completely by previous work (Hughes et al., 2015) for the HDP topic model. For all equations from that paper, you need only substitute in $K+1$ for the number of documents $D$ to translate from topic models to HMMs.

To begin our derivation, we rewrite the overall (surrogate) objective function $\mathcal{L}$ as a function of $\hat{\rho}$ and $\hat{\omega}$, the parameters the global step updates. This means regrouping fragments of the surrogate term $\mathcal{L}_{\text{sur}}$, the top-level term $\mathcal{L}_{\text{hdp-global}}$, and the subset of $\mathcal{L}_{\text{hdplocaltrans}}$ that depends on $q(u)$. Combining all these terms together and substituting in the surrogate cumulant bound gives the complete objective necessary for learning $\hat{\rho}, \hat{\omega}$. Note that we have dropped any additive terms constant with respect to $\hat{\rho}, \hat{\omega}$ in this expression, since they have no bearing for our numerical optimization problem.

$$\mathcal{L}_{obj}(\hat{\rho}, \hat{\omega}) = \sum_{k=1}^{K} \Bigg( -c_B(\hat{\rho}_k \hat{\omega}_k, (1 - \hat{\rho}_k)\hat{\omega}_k) \quad (40)$$

$$+ \Big( K + 1 - \hat{\rho}_k \hat{\omega}_k \Big)\Big( \psi(\hat{\rho}_k \hat{\omega}_k) - \psi(\hat{\omega}_k) \Big)$$

$$+ \Big( K(K+1-k) + 1 + \gamma - (1-\hat{\rho}_k)\hat{\omega}_k \Big)\Big( \psi((1-\hat{\rho}_k)\hat{\omega}_k) - \psi(\hat{\omega}_k) \Big) \Bigg)$$

$$+ \sum_{\ell=1}^{K} \mathbb{E}_q[\beta_\ell] \Bigg( \log(\alpha + \kappa) - \log \kappa + \sum_{k=0}^{K} \alpha_k P_{k\ell}(\hat{\theta}) \Bigg)$$

$$+ \mathbb{E}_q[\beta_{K+1}] \Bigg( \sum_{k=0}^{K} \alpha_k P_{k,K+1}(\hat{\theta}) \Bigg)$$

where $\alpha_k = \alpha$ for $k \geq 1$.

### E.1 Constrained Optimization Problem

Our goal is to find the $\hat{\rho}, \hat{\omega}$ that maximize $\mathcal{L}_{obj}$. Remember that $\hat{\rho}, \hat{\omega}$ parameterize $K$ Beta distributions, and so have certain positivity constraints. Thus, we need to solve a *constrained* optimization problem:

$$\text{argmax}_{\hat{\rho}, \hat{\omega}} \quad \mathcal{L}_{obj}(\hat{\rho}, \hat{\omega}) \quad (41)$$

$$\text{subject to} \quad \hat{\rho}_k \in (0, 1) \quad \hat{\omega}_k > 0, \quad k = 1, \ldots, K$$

We now give the expressions for the gradient $\nabla \mathcal{L}_{obj}$, with respect to each entry of $\hat{\omega}$ and $\hat{\rho}$.

**Gradient for $\hat{\omega}$.** Taking the derivative of Eq. (40) with respect to each entry $\hat{\omega}_m$ of $\hat{\omega}$, for $m \in 1, 2, \ldots K$, is:

$$\frac{\partial \mathcal{L}_{obj}}{\partial \hat{\omega}_m} = \Big( K + 1 - \hat{\rho}_m \hat{\omega}_m \Big)\Big( \hat{\rho}_m \psi_1(\hat{\rho}_m \hat{\omega}_m) - \psi_1(\hat{\omega}_m) \Big) \tag{42}$$
$$+ \Big( K(K + 1 - m) + 1 + \gamma - (1 - \hat{\rho}_m)\hat{\omega}_m \Big)\Big( (1 - \hat{\rho}_m)\psi_1((1 - \hat{\rho}_m)\hat{\omega}_m) - \psi_1(\hat{\omega}_m) \Big)$$

where $\psi_1 \triangleq \frac{d^2}{dx^2} \log \Gamma(x)$ is the trigamma function.

**Gradient for $\hat{\rho}$.** Define $\Delta$ as a $K \times K + 1$ matrix of partial derivatives of $\mathbb{E}_q[\beta_k]$:

$$\Delta_{mk} \triangleq \frac{\partial}{\partial \hat{\rho}_m} \mathbb{E}_q[\beta_k] = \begin{cases} -\frac{1}{1-\hat{\rho}_m} \mathbb{E}_q[\beta_k] & m < k \\ \frac{1}{\hat{\rho}_m} \mathbb{E}_q[\beta_k] & m = k \\ 0 & m > k \end{cases} \tag{43}$$

Now, the derivative of $\mathcal{L}_{obj}$ with respect to each entry $\hat{\rho}_m$ of the vector $\hat{\rho}$, for $m \in 1, 2, \ldots K$, is:

$$\frac{\partial \mathcal{L}_{obj}}{\partial \hat{\rho}_m} = \hat{\omega}_m \Big( K + 1 - \hat{\rho}_m \hat{\omega}_m \Big)\psi_1(\hat{\rho}_m \hat{\omega}_m) \tag{44}$$
$$- \hat{\omega}_m \Big( K(K + 1 - m) + 1 + \gamma - (1 - \hat{\rho}_m)\hat{\omega}_m \Big)\psi_1((1 - \hat{\rho}_m)\hat{\omega}_m)$$
$$+ \sum_{\ell=1}^{K} \Delta_{m\ell} \Big( \log(\alpha + \kappa) - \log \kappa + \sum_{k=0}^{K} \alpha_k P_{k\ell}(\hat{\theta}) \Big)$$
$$+ \Delta_{m,K+1} \Big( \sum_{k=0}^{K} \alpha_k P_{k,K+1}(\hat{\theta}) \Big)$$

## E.2 Unconstrained Optimization Problem

In practice, we find that it is numerically more efficient to first transform our constrained optimization problem above into an unconstrained problem, and then solve the unconstrained problem via a modern gradient descent algorithm (L-BFGS).

Both target variables $\rho, \omega$ have simple bound constraints on each of their $K$ entries. Each entry of $\rho$ lies in $[0, 1]$, while each entry of $\omega$ must be larger than 0. We can define an invertible transform between constrained scalars $\rho_k, \omega_k$ and unconstrained real scalar variables $c_k, d_k$ as follows:

$$c_k \triangleq \text{sigmoid}^{-1}(\rho_k) \qquad\qquad \rho_k \triangleq \text{sigmoid}(c_k) = \frac{1}{1 + e^{-c_k}} \tag{45}$$
$$d_k \triangleq \log \omega_k \qquad\qquad \omega_k \triangleq e^{d_k}$$

As shorthand, we write $\rho(c)$ to denote the vector $\rho$ obtained by transforming the input vector $c$. Similarly, we write $\omega(d)$ to be the vector $\omega$ obtained by applying the transform to input $d$. We can then define an unconstrained optimization problem

$$c^*, d^* \leftarrow \text{argmax}_{c,d} \ \mathcal{L}_G\Big(\rho(c), \omega(d)\Big) \tag{46}$$

The optimal values $c^*, d^*$ can be simply transformed to $\rho^*, \omega^*$, which are by construction optimal solutions to our original problem.

Our unconstrained objective can be solved via gradient descent, where the gradients can be easily computed by the chain rule using our original gradients with respect to $\rho, \omega$ as inputs.

The gradient at entry $m$ of vector $c$ is

$$\frac{\delta}{\delta c_m}[\mathcal{L}_G] \triangleq \frac{\delta}{\delta c_m}[\rho_m] \cdot \frac{\delta}{\delta \rho_m}\mathcal{L}_G \qquad (47)$$

$$= \rho_m(1 - \rho_m)\frac{\delta}{\delta \rho_m}\mathcal{L}_G, \quad \text{where } \rho_m \triangleq \frac{1}{1 + e^{-c_m}}$$

Similarly, the gradient at entry $m$ of vector $d$ is

$$\frac{\delta}{\delta d_m}[\mathcal{L}_G] \triangleq \frac{\delta}{\delta d_m}[\omega_m] \cdot \frac{\delta}{\delta \omega_m}\mathcal{L}_G \qquad (48)$$

$$= \omega_m\frac{\delta}{\delta \omega_m}\mathcal{L}_G, \quad \text{where } \omega_m \triangleq e^{d_m}$$

## F. Merge move details.

As in previous work, our merge moves propose a candidate model in which all posterior assignment mass from two original states $j, k$ has been combined into a new single state $m$ that replaces both $j, k$. This proposed candidate model is evaluated under the variational objective function $\mathcal{L}$. If the new objective value improves on (has larger value than) the original, we keep the candidate. Otherwise we discard it and continue with the original.

### F.1 Construction of candidate proposal

Below, we outline the four stages of constructing the candidate proposal: (1) selecting the pair of states to try to merge, (2) creating candidate local free parameter values $s'$, which are interpreted as soft assignment state sequences; (3) creating the representative sufficient statistics $N', M'$, and (4) creating candidate global parameters $\rho', \omega', \pi', \tau'$. Note that step 2 is a purely "conceptual" step: we need not explicitly construct local parameters $s'$ of the candidate in the implementation. Instead, we can directly manipulate sufficient statistics to avoid huge runtime and memory costs of local construction (e.g. when there are hundreds of sequences).

In the last subsection, we'll discuss how to evaluate the objective function for the candidate configuration. Throughout the process, remember that the input for a proposal is the current model, represented by the current free variational parameters $\hat{s}, \hat{\rho}, \hat{\omega}, \hat{\pi}, \hat{\tau}$.

### Proposal Step 1/4: State selection.

Selecting a promising pair of states to merge is critical. We can just propose a pair at random, but this is unlikely to succeed. Instead, we can rate all possible pairs based on the partial objective

$$\mathcal{L}'_{partial} \triangleq \mathcal{L}'_{\text{data}}(x, S', \hat{\tau}') + \mathcal{L}_{\text{hdplocaltrans}}(M', \hat{\rho}', \hat{\theta}') + \mathcal{L}_{\text{sur}}(\hat{\rho}', \hat{\omega}') + \mathcal{L}_{\text{hdp-global}}(\hat{\rho}', \hat{\omega}') \qquad (49)$$

where each candidate quantity is quickly computed via the procedures defined below. Any pairs for which this score is below zero need not be considered, because $\mathcal{L}'_{partial} < 0$ implies that $\mathcal{L}' < \mathcal{L}$.

### Proposal Step 2/4: Local parameter construction

To construct the candidate, we assume a single fixed rule for creating $s'$ from $s$, the originally inferred state sequence probabilities. The rule is simply this: any mass previously assigned to states $j, k$ is now assigned to a new state, $m$.

To illustrate how to create $s'$ from $s$, we'll first show a few examples. We'll assume that the dataset $x$ has only one sequence (thus dropping the $n$-index), and that our original model has $K = 4$ states.

We'll consider a slice of $s$ at a particular timestep $t$. This is a $K \times K$ matrix, where entry in row $j$ and column $k$ gives the probability that $z_{t-1}$ is in state $j$ and $z_t$ is in state $k$.

$$s_t = \begin{bmatrix} .01 & .02 & .03 & .14 \\ .01 & .12 & .13 & .04 \\ .01 & .12 & .13 & .04 \\ .11 & .02 & .03 & .04 \end{bmatrix} \tag{50}$$

First, suppose we have selected to merge states 3 and 4. Below is the resulting candidate $s'$.

$$s_t = \begin{bmatrix} .01 & .02 & .03 & .14 \\ .01 & .12 & .13 & .04 \\ .01 & .12 & .13 & .04 \\ .11 & .02 & .03 & .04 \end{bmatrix} \rightarrow \text{Merge 3\&4} \rightarrow \quad s'_t = \begin{bmatrix} .01 & .02 & .17 \\ .01 & .12 & .17 \\ .12 & .14 & .24 \end{bmatrix} \tag{51}$$

Alternatively, suppose we are merging states 1 and 2. The resulting $s'$ is

$$s_t = \begin{bmatrix} .01 & .02 & .03 & .14 \\ .01 & .12 & .13 & .04 \\ .01 & .12 & .13 & .04 \\ .11 & .02 & .03 & .04 \end{bmatrix} \rightarrow \text{Merge 1\&2} \rightarrow \quad s'_t = \begin{bmatrix} .16 & .16 & .18 \\ .13 & .13 & .04 \\ .13 & .03 & .04 \end{bmatrix} \tag{52}$$

Generalizing from these two examples, we can formalize the construction rules. First, some definitions. We can divide the original states into two mutually exclusive sets: $P$ and $\bar{P}$. Let $P = \{j, k\}$ denote the pair of states being merged and $\bar{P}$ the set of all other states. Then, we can give formulas for every entry of $s'$ as follows:

$$
\begin{aligned}
s'_{tmm} &= s_{tjj} + s_{tkk} + s_{tjk} + s_{tkj} \\
s'_{tm\ell} &= s_{tj\ell} + s_{tk\ell}, \quad \forall \ell \in \bar{P} \\
s'_{t\ell m} &= s_{t\ell j} + s_{t\ell k}, \quad \forall \ell \in \bar{P} \\
s'_{t\ell\ell'} &= s_{t\ell\ell'}, \quad \forall \ell, \ell' \in \bar{P}
\end{aligned}
\tag{53}
$$

Here, for simplicity, our indexing system ignores the fact that the matrix will get re-indexed after the merge. That is, if we merge states 1 and 2, then what was originally state 4 will become state 3 in the candidate.

When applied at every timestep $t$, this procedure will output a $s'$ with $K-1$ states. By construction, we know that the output $s'$ is a valid "soft" state sequence, meaning it satisfies the requirements of positivity and sums-to-unity:

$$s'_{tjk} > 0 \qquad \sum_{j=1}^{K-1} \sum_{k=1}^{K-1} s'_{tjk} = 1 \quad \forall t \in \{1, 2, \dots T-1\} \tag{54}$$

Thus, we can use $s'$ as a valid free parameter for the candidate variational distribution $q'(z)$. Note that given $s'_t$ (adjacent timestep joint distribution), we can easily construct the parameters $r'$ (the single-timestep marginals), and $\sigma'_t$ (adjacent-timestep conditionals) using their definitions. Thus, our local parameter construction is complete.

### Proposal step 3/4: Sufficient statistics

In this proposal step, we need a procedure to summarize the new soft state sequence assignments $s'$ into the sufficient statistics $M'$ (state transition counts) and $S'$ (data statistics). Naturally, we can simply follow the definitions of these statistics, computing the values of $M', S'$ via relevant sums over instantiations of $s', r'$. However, instantiating the local parameters and computing the sum can be expensive (linear in the number of sequences $N$ and timesteps $T$) in runtime and memory.

**Fast computation from original statistics.** Alternatively, we can avoid summing over $s', r'$ and instead directly manipulate the original values of $M$ and $S$ to determine $M', S'$. Given the additive construction of $s'$ from $s$, it is easy to show the same rules apply to manipulate sufficient statistics.

First, for data statistics $S$ we have a simple rule:

$$S'_m = S_j + S_k, \qquad S'_\ell = S_\ell, \ell \in \bar{P} \tag{55}$$

Next, for state transition counts $M$ we can follow the same rules as for constructing $s'_t$:

$$
\begin{aligned}
M'_{mm} &= M_{jj} + M_{kk} + M_{jk} + M_{kj} \\
M'_{m\ell} &= M_{j\ell} + M_{k\ell}, \quad \forall \ell \in \bar{P} \\
M'_{\ell m} &= M_{\ell j} + M_{\ell k}, \quad \forall \ell \in \bar{P} \\
M'_{\ell\ell'} &= M_{\ell\ell'}, \quad \forall \ell, \ell' \in \bar{P}
\end{aligned} \tag{56}
$$

The resulting values of $M', S'$ from these rules will exactly represent the whole dataset under candidate local assignments $s', r'$. Thus, constructing $M', S'$ from $M, S$ is exact yet much more affordable than naive construction via sums of the local parameters.

### Proposal Step 4/4: Global parameter construction

The conclusion of step 3 above yields valid summaries $M', S'$ for a candidate model with $K - 1$ states. We now develop the procedure for creating candidate global parameters.

**Data-generation parameters**. To construct $\{\hat{\tau}'_k\}_{k=1}^{K-1}$, we can simply apply the closed-form global update with the candidate summaries $\{S'_k\}_{k=1}^{K-1}$. Note that any unaffected state in set $\bar{P}$ need not be edited at all.

**HDP state-appearance probability parameters** Given summaries $M'$ for a candidate with $K - 1$ states, there is not a straight-forward way to simultaneously construct $\hat{\theta}$ (which control the state-transition variational factor on $\pi_j$) and $\hat{\rho}, \hat{\omega}$ (which control the top-level conditional probabilities $u$). The variational update to $\hat{\theta}$ depends on $\rho$, and vice versa.

To escape this "chicken-and-egg" problem, we propose the following step-by-step procedure, which uses our original parameters $\hat{\rho}$ to estimate an initial candidate $\hat{\rho}'$, and then follows-up with a conventional update to $\hat{\theta}'$.

First, we obtain $\hat{\rho}'$ (vector of size $K - 1$) via a series of closed-form arithmetic operations:

- Compute $\beta_\ell^* \triangleq \mathbb{E}_{q(u|\rho)}[\beta_\ell]$ for every original state $\ell \in \bar{P} \bigcup \{j, k\}$

- Compute $\beta_m^* = \beta_j^* + \beta_k^*$.

- Calculate $\rho'$ that would satisfy $\mathbb{E}_{q(u|\rho')}[\beta_\ell] = \beta_\ell^*$, for every candidate state $\ell \in \bar{P} \bigcup \{m\}$

Finally, given $\hat{\rho}'$, and the summary $M'$, we update $\hat{\theta}'$ via the closed-form update rule in Sec. 3 of the main paper. Recall that this rule does not involve $\hat{\omega}'$ at all. Optionally, we can perform one final update to $\hat{\rho}', \hat{\omega}'$ given the new $\hat{\theta}'$. This is recommended. Alternatively, we can keep the temporary value of $\hat{\rho}'$ defined above and obtain a heuristic value for $\hat{\omega}'$ (vector of size $K - 1$) from the original by simply keeping all original entries of $\hat{\omega}$ that are not involved in the merge as they are, and setting the merged state $m$'s to $\hat{\omega}'_m = \hat{\omega}_j + \hat{\omega}_k$.

### F.2 Evaluation of $\mathcal{L}$ for candidate merges

Given a candidate set of free parameters for a model with $K - 1$ states, we need to evaluate the objective $\mathcal{L}(\hat{s}', \hat{\theta}', \hat{\rho}', \hat{\omega}', \hat{\tau}')$ to decide if it improves on the original objective score.

Examining the formula for $\mathcal{L}$, we can see that global parameters $\hat{\theta}', \hat{\rho}', \hat{\omega}', \hat{\tau}'$ and sufficient statistics $M', S'$ allow computation of all terms but one: the term $\mathcal{L}_{\text{entropy}}$. Otherwise, we could construct a candidate model *and* make a decision about acceptance without touching any local data.

However, this entropy term must be calculated by explicitly instantiating $s'_t$ at every timestep. Note that we need not instantiate all slices simultaneously. Instead, we can take the original value of $s$, and iterate over timesteps, summing up the relevant data to compute $\mathcal{L}_{\text{entropy}}(\hat{s}')$ as we go, lowering memory usage from $\mathcal{O}(TK^2)$ to $\mathcal{O}(K^2)$.

### F.3 Attempting multiple non-overlapping pair-wise merges at once.

The merge procedure above outlines how to construct and evaluate a candidate model for a single merge pair. However, examining only one pair at a time may require many passes through a large dataset before all relevant merges are attempted and accepted. The natural question is whether we can simultaneously evaluate and accept several merge pairs in a single pass through the dataset, as is possible with DP mixtures in Hughes and Sudderth (2013) and HDP topic models in Hughes et al. (2015).

The only problematic term in this regard is the entropy, $\mathcal{L}_{\text{entropy}}$. All other terms in the objective are either purely functions of global variables or linear functions of sufficient statistics. Earlier, Eq. (8) decomposes $\mathcal{L}_{\text{entropy}}$ into a sum over the entries of a non-negative matrix $H$. As above, we'll assume there is only one sequence for simplicity, so the notation below will omit the index $n$ which specifies which sequence is under consideration. First, we define special starting-state entropy term $H_{0k}$:

$$H_{0k} \triangleq -\hat{r}_{1k} \log \hat{r}_{1k} \tag{57}$$

Next, we define the state-transition entropy $H_{jk}$ for $1 \leq j \leq K$ as:

$$H_{jk} \triangleq -\sum_{t=2} \hat{s}_{tjk} \log \frac{\hat{s}_{tjk}}{\hat{r}_{tj}} \tag{58}$$

Using these positive scalars $H_{jk}$, which always satisfy $H_{jk} > 0$ for all values $0 \leq j \leq K$ and $1 \leq k \leq K$, we can rewrite the entropy term $\mathcal{L}_{\text{entropy}}$ as a sum over entries of $H$:

$$\mathcal{L}_{\text{entropy}} = \sum_{k=1}^{K} \sum_{j=0}^{K} H_{jk} \tag{59}$$

**Single pair computation.** First, consider how we should efficiently compute the entropy for a single merge pair $j, k$, given $s'$. From the construction of $s'$ in Eq. (53), we see that for large $K$, any entries in the entropy table $H$ which have both states not involved in the merge will not change at all. That is, for $\ell, \ell' \in \bar{P}$, both $s'_{\ell,\ell'} = s_{\ell,\ell'}$ and $\sigma'_{\ell,\ell'} = \sigma_{\ell,\ell'}$. Thus, $H'_{\ell,\ell'} = H_{\ell,\ell'}$, and we need not recompute this value.

The values that will change are any entry of $H$ that involve either $j$ alone, $k$ alone, or both $j$ and $k$. There are exactly $2K - 3$ such entries. So, we need to compute only $2K - 3$ entries of $H'$ from-scratch. The rest we can copy from the original matrix $H$ directly into the appropriate entries of $H'$.

What is the computational benefit to this procedure? If we naively computed the complete scalar entropy term for a merge pair, we would require traversing $T$ timesteps, and perform $K^2$ work at each one. Using the scheme described above, we perform only $\mathcal{O}(TK)$ work to compute a $H'$ table once given the original $H$.

**Multiple pair computation** Now, consider two candidate merge pairs: $j, k$ into new state $m$ and $j', k'$ into new state $m'$. We could of course consider an either-or decision, where one of these candidates is accepted but not both.

However, we wish to contemplate accepting both. Specifically, the case that both $j = j'$ or $k = k'$ is definitely not going to work. There would be substantial overlap among the $2K - 3$ "replacement" entries of $H'$ that need to be computed for each set.

The case where $j, k, j'$, and $k'$ are all distinct is more manageable, but still we will not be able to evaluate $\mathcal{L}''_{\text{entropy}}$ (the objective if both were accepted) using only the $2K - 3$ entries of $H'$ outlined above. There will be two entries of these "substitution" values $H'$ that will be incorrect. The specific two entries that *must* be custom-computed are $H'_{mm'}$ and $H'_{m'm}$, which will involve the transitions from new state $m$ from the first pair to new state $m'$ of the second, and vice versa. We cannot exactly evaluate $\mathcal{L}''_{\text{entropy}}$ without considering these terms.

However, we can consider a *pessimistic* estimate of $\mathcal{L}_{\text{entropy}}$ for the case of a double merge. As entropies are always non-negative, we can easily leave out $H'_{mm'}$ and $H'_{m'm}$ from the sum in Eq. (59). From the basic properties of entropy, we know that the entropy term of a candidate will always decrease from the original, so this omission provides a rather tight lower bound on the candidate entropy. In practice, we find this "omit entries" strategy is enough to guarantee acceptance on many real-world datasets.

## G. Delete move details.

### G.1 Candidate selection.

Deletes are more flexible than merges at reassigning mass but require expensive local steps on the target data. To keep costs affordable, before each lap we find a group of $L$ states whose unified target set is at most 10 sequences. We prioritize states that have not been attempted before. We then collect the unified target set and try all $L$ deletes one at a time. If no group can be found ($L = 0$), no move is performed.

### G.2 Delay of delete proposals.

Given a naive initialization, greedily applying many delete proposals may yield short-term improvements in the objective $\mathcal{L}$, but long-term under-performance. This can occur because given enough update-cycles, a few junk states may evolve into useful states that the model prefers not to delete. This is especially important if birth moves are not enabled, which means the model cannot ever expand its truncation after a deletion. When possible, we recommend *delaying* the onset of delete proposals for several complete laps. In our experiments, we found that delays of 5 laps are sufficient.

## H. Birth move details.

### H.1 Proposal selection details.

**Bisection proposal: Linear search that improves $\mathcal{L}_{\textbf{data}}$** We recommend one concrete proposal that takes advantage of sticky state persistence: First, choose an interval $[a, b]$ of the current estimated sequence $z_{n1}...z_{nT}$, where we set $z_{nt} = \arg\max_k[\hat{r}_{n1} \ \hat{r}_{n2} \ ... \ \hat{r}_{nK}]$, which is its max-marginal assignment. We choose end points $[a, b]$ randomly based on the current changepoints of sequence $z_n$. We sometimes randomly perturb these endpoints to look at intervals that are not necessarily current segment boundaries.

Given interval $[a, b]$, the goal is to split this interval into two contiguous blocks each assigned to a new state, or just produce one state for the whole block if that is more favorable. To do so, we perform a linear search for an optimal cut point $m$ satisfying $a < m < b$. The search objective is to maximize the term $\mathcal{L}_{\text{data}}$ for the two potential state statistics $S_{am}, S_{bm}$, where $S_{am}$ represents the statistics of $x_n$ over the timestep interval $[a, m]$, and likewise $S_{mb}$ covers the interval $[m, b]$. Concretely, $S_{am} = \sum_{t=a}^{m} s_F(x_{nt})$, and $S_{mb} = \sum_{t=m+1}^{b}$. The score we evaluate simplifies from the general $\mathcal{L}_{\text{data}}$ term to:

$$\mathcal{L}_{\text{data}}(S_{am}, S_{mb}) = -c_H(S_{am} + \bar{\tau}) - c_H(S_{mb} + \bar{\tau}) + \text{const} \tag{60}$$

This search for $m$ works very well in practice and takes only linear time (in length of the interval) to evaluate. To control cost on very long intervals, we may only consider a subset of possible $m$ values evenly spaced within $[a, b]$.

This procedure is enabled in the bnpy software by the keyword option `initname=bisectGrownBlocks`. See the plain-text file `settings-bnpyHDPHMMcreateanddestroy.txt` for where this is established within the `x-hdphmm-nips2015` repository.

**Smart selection of intervals to target.**   With large models, naively choosing intervals or states to target will often waste time on proposals that are unlikely to be accepted. As a simple remedy, we recommend tracking an integer count of how many previous failures have occured with each existing state, and biasing the selection of intervals $[a, b]$ within $z_n$ towards either those that belong to a state which has yielded successful proposals in the recent past, or a state that have not been targeted recently. Alternatively, more data-driven selection is possible and would likely be effective.

### H.2  Proposal evaluation details.

**Proposal Refinement.**   Given proposed new local parameters $\hat{r}'_n, \hat{s}'_n$ for the sequence $n$ from our bisection procedure, we refine these values by running several iterations of memoized local and global updates. The final output of the refinement phase is the pair of revised values $\hat{r}'_n, \hat{s}'_n$. The cost of each refinement iteration is dominated by the $\mathcal{O}(T_n K'^2)$ dynamic programming required for the local step. So long as $K'$ does not get too large, this remains very affordable. The advantage is that we can greatly improve the chances of success. In practice, each time we visit a sequence $n$ we perform several proposals (each followed by one or more refinement steps), and then accept or reject the whole new configuration once. We find this is a quick way to make large changes.

**Batches with multiple sequences.**   If batches have multiple sequences, bookkeeping becomes slightly more involved but births guaranteed to improve the objective are still possible. We first illustrate the case of two sequences in the batch, labelled $n_1$ and $n_2$. When we visit the first sequence $n_1$ for a local step, we have statistics $S, M$ representative of the whole dataset (including the previous assignments to $n_1$ and $n_2$), and previously stored batch statistics $S_b, M_b$, which summarize both $n_1$ and $n_2$. When we propose new candidate assignments for $n_1$, we need to allow new states to also propagate to $n_2$. To do so, we use candidate statistics $M', S'$ formed by including both the previous stats $M_b, S_b$ for the batch and the new assignments to $n_1$.

$$S' = S + S'_{n_1} \quad M' = M + M'_{n_1} \tag{61}$$

While this construction technically double-counts sequence $n_1$, these statistics are temporary and used only to create temporary global parameters the proposal of $n_2$. This allows $n_2$ to use any states it used previously (since its old assignments are counted) and any new states found in $n_1$ (since those new assignments are counted). It's vitally important to keep the previous batch assignment statistics, otherwise if $n_2$ had previously used some unique states found in no other sequence, candidate states $S', M'$ without those would lead to global parameters without them and a local step where they would effectively disappear from $n_2$.

For batches with many sequences, we keep up the successive process of proposing some new states at sequence $n$, and then retaining these new assignments (and any other new assignments made) as well as previous assingments in the temporary statistics. Again, we emphasize that this double-counting is temporary, and only used to find a final set of proposed values $\{\hat{r}_n, \hat{s}_n\}$ for all sequences $n$ in the batch $b$. After proposing for all sequences in the batch, the accept or reject decision is made with exact statistics that reflect the new assignments only (and fixed assignments from other batches).