[Reviews · NeurIPS 2015]

Submitted by Assigned_Reviewer_1

Fig 1: identify x and y axes.

Deletes -> Deletions.

Thru -> Through
Summary: This paper proposes several contributions to variational inference in sticky HDP-HMM models. The paper is well written and contains worthy technical improvements on the state-of-the-art. I only have some very minor comments:

Submitted by Assigned_Reviewer_2

In their paper the authors propose a variational inference algorithm for HDP-HMM. Similar to a sampler this method includes birth, merge, and delete steps, so that the maximum number of states can be adapted to the complexity visible in the observations, but only updates improving the fit are accepted. Scalability to large data sets is achieved by dividing them into batches and sharing only sufficient statistics and global parameters. Experiments indicate that this approach works well and can be parallelized successfully.

The paper is well written and explains the non-parametric model as well as the proposed algorithm very clearly. Using steps from MCMC methods in variational inference is an interesting idea, which could be applied to other non-parametric models, too. As inference combining large data sets with advanced models is an active research topic, this paper is really relevant.

However, the results of the baseline algorithm for the simple experiment shown in figure 3 seem to be strange and the rebuttal did not convince me that that is the typical behavior of standard algorithms on easy data sets.
Summary: Paper presents a novel and efficient variational inference algorithm for HDP-HMM with large data sets based on the inventive approach of using MCMC steps to adapt number of states.

Submitted by Assigned_Reviewer_3

This paper applies to the sticky HDP-HMM some recently developed machinery for split/merge style adaptation of the truncation level for variational inference in Bayesian nonparametric models and evaluates the resulting algorithm on several tasks, emphasizing identifying the true number of states present in a dataset. The algorithm is an improvement over the algorithm presented in [7] particularly because of these extra steps, and possibly due to the improved treatment of the variational factor on the top-level HDP weights.

The paper is written very clearly. However, given previous work on these subjects the novel contributions here are incremental. Furthermore, the experiments are unenlightening and the synthetic experiment is very confusing.

## Experiments ## The synthetic data results presented in the first experiment don't look correct. The data, which I understand are shown in the top-left panel, are trivially separable and any algorithm, even a GMM without any temporal dynamics let alone sticky bias, would separate these clusters perfectly in just a few iterations (unless perhaps it had some adversarial initialization). Yet the reported experiment shows that a sampler (presumably operating on the entire dataset in each iteration) required 1000 iterations just to find these obvious clusters? Or that SVI (without split/merge moves) couldn't get away from modeling the data with 60 clusters after 1000 full passes through the data? These results contradict both reported performance in the literature and my own experience with related models, even vanilla GMMs.

The other experiments are believable but they don't provide much insight into the potential tradeoffs here or failure modes of the proposed algorithm; each figure just serves to show that the proposed method always works great. Studying Hamming error and the number of recovered states is okay, but those metrics probably favor the proposed method's emphasis on searching explicitly over the model size; why not show predictive likelihood as well to measure whether the predictive distributions are converging for these other methods (as expected)? When does this method do badly, and when might its extra computational burden not be worthwhile? Instead of shedding light on these kinds of questions, these experiments are all salesmanship.

## Originality / significance ## The originality of this work relative to [13] and [14] is minimal. The authors helpfully enumerate contributions over [14] in the conclusion section. Three of the four seem incremental while the last (an implementation with parallelism) isn't relevant to deciding the merits of the paper.

The extension from the HDP-HMM to the Sticky HDP-HMM is pretty much the first thing to try: the sticky bias amounts to just adding an extra weight to a particular entry in each second-level weight vector in the HDP, and so the extent of the contribution here is just using the bound from [14] and splitting a logarithm over a sum (Eq. (36) in the supplement). (The algebra would be clearer if it were written as log(alpha beta_k + kappa) = log(beta (alpha + kappa) + (1-beta) kappa) >= beta log(alpha+kappa) + (1-beta) log(kappa).)

The merge selection and delete acceptance rules are an improvement over [14], but they simply amount to a better heuristic. The birth moves also seem like a reasonable choice but the idea is pretty straightforward given the use of birth moves in [13].

Parallelization would be a merit of the implementation, but not the method as described in the paper, especially since all the comparable methods admit the same (easy) parallel implementations. If we're meant to evaluate the quality of the implementation as a contribution of this paper, the implementation should be included.

## Misc ## - The comments on model selection don't make sense (Lines 049 and 174). Are these comments meant to refer to using the variational lower bound on the model evidence to compare the HDP-HMM to other models? If so, it's not been made clear that these bounds for these models are useful proxies compared to direct estimates via AIS or SIS, and if that's a claimed merit of this approach, it should be investigated in the experiments. The comment on Line 174 that bounds which use a delta factor on the top-level weights "can favor models with redundant states" is not demonstrated; indeed, those do provide valid bounds on model evidence with just the top-level weights fixed instead in precisely the same way that the each majorization bound used in the EM algorithm is a bound on the likelihood of the data with the hidden variables marginalized out (but the parameters fixed). - Around line 236 it is claimed that memoized variational inference with one pass (one lap) is identical to SVB. Is that essentially a choice to use SVB to start the algorithm as opposed to initializing the cached statistics (M, S)? Since [13] was published at the same time as [20] and does not cite [20], it's surprising to learn that memoized variational inference actually contains streaming variational inference as a special case. - Is memoized variational inference just regular batch mean field with a different update schedule? Caching sufficient statistics, instead of keeping around individual factors' parameters only to throw them out when they are updated, seems to be the straightforward implementation strategy for such an update schedule. - In supplement C.1, a line reads "Here, all the infinite sums are convergent. This allows some regrouping" but infinite sums need to be *absolutely* convergent to allow regrouping. If all terms are nonnegative, regrouping is always okay. If the series is convergent but with indefinite terms, regrouping does not hold in general.
Summary: The paper is very clearly written and by combining ideas in the literature demonstrates a fine, if complicated, inference algorithm. However, given the past body of work on the subject, the contributions here are not significant, and the experimental evaluation, while diverse, provides little insight and includes a synthetic experiment that looks incorrect.

Author Feedback
Author rebuttal: We develop scalable variational for the sticky HDP-HMM, using birth/merge/delete moves to add or remove states and better fit data. Previous methods like stochastic variational (Johnson et al 2014) or blocked Gibbs samplers (Fox et al 2011) cannot add new clusters after initialization, and cannot remove junk states effectively. A primary goal of our work is to show that these fixed-K algorithms converge slowly and get stuck easily on even modest datasets, while our method finds better segmentations faster and more consistently. We will release a polished Python implementation so others can try our method easily and repeat any experiment themselves.

## R2: toy data results in Fig. 3 don't look correct

We included this figure to raise this suprising point: even on apparently "easy" datasets, methods like Gibbs sampling and stochastic variational can take unacceptably many iterations to reach ideal segmentations. For this experiment, we initialize all methods with 50 or 100 clusters, each cluster created using a random contiguous window of data. This initialization inevitably contains many redundant clusters, which baseline methods cannot prune easily since they lack merge moves and must wait for one member of a redundant set to overpower the others.

Several papers warn about slow mixing, including Fig. 15 of Fox et at 2011 (where "convergence" takes 100k iterations on just 1 sequence) and classic work by Jain and Neal on split-merge MCMC. We expect this "contradicts" earlier work only because many toy experiments are very small in scale or use initial K values close to the true number of clusters, which can be unrealistically optimistic. We emphasize that we used Fox's sampler code out-of-the-box, only changing initialization and hyperparameter settings to match other methods exactly.

## R2: birth moves "incremental" given births for DP mixtures in Hughes, Sudderth '13

There are some exciting new features of our proposed births for HMMs. First, the linear-time cut-point search driven by the objective function term L_data. Second, these births are guaranteed to improve the whole-dataset objective, while those in H&S '13 have no checks in place and can result in worse ELBO scores.

## R2: merge, delete moves are "incremental" given prior work

While merge and delete moves have the same goals as in other work, implementations are model-specific and can require critical design choices to be effective. For example, unlike DP mixtures and HDP topic models, the HDPHMM has some statistics that involve interacting pairs of states. Developing methods that can accurately construct these O(K^2) statistics and evaluate the ELBO objective after several accepted merge pairs, one-after-the-other, required careful innovation (see lines 308-311 and supplement).

## R2: experiments lack predictive likelihood as a metric

We chose to measure number of states and Hamming distance over predictive likelihood because they capture a key goal of unsupervised learning: compact, interpretable models. The basic Gibbs sampler can often reach decent predictive performance while having redundant copies of some states/clusters. For a biologist trying to understand a set of chromatin states, the presence of junky states is a huge issue: are they really preferred or an artifact of slow-converging inference? Redundancy also reduces scalability to big datasets. Baselines don't provide an effective answer to this question, but our method does. That said, in a future revision we will include a predictive likelihood plot for at least some datasets, to help readers understand tradeoffs between these metrics.

## R2: when does the method do badly? when is it not worth the extra cost?

We will revise with more candid discussion of weaknesses. In particular, runs of the method are rarely perfect (see purple lines converging to different plateaus in Fig 5), though typical runs are usually better than baselines. Also, the birth uses only data from one sequence, so any cluster that cannot be justified from one sequence will be missed. Nevertheless, we find that local optima issues with baselines are much worse.

## R2: comments on model selection dont make sense

The point estimate case requires MAP learning of q(beta), which means the overall objective is actually a bound on p(data, beta), not p(data). As K increases we can always improve this objective, and because beta is infinite-dimensional for the HDPHMM, it can be increased without bound. Only a variational treatment of q(beta) with variance > 0 provides regularization to penalize extra empty topics. There are experiments to this effect in Fig. 2 of Hughes, Kim, Sudderth 2015, which we referenced but can reproduce for our model/bound in the supplement.

## R2 misc: suprising that Streaming variational [20] equivalent to memoized

This fact is acknowledged in [20] itself: their Alg. 3 is a one-pass memoized (aka sufficient statistic update) algorithm.